# Pan-tissue analysis of allelic alternative polyadenylation suggests widespread functional regulation

Yisheng Li[1,2,§], Bernhard Schaefke[1,3,§,*], Xudong Zou[1], Min Zhang[1], Florian Heyd[2], Wei Sun[1,†], Bin Zhang[1,‡], Guipeng Li[1,3], Weizheng Liang[1], Yuhao He[1], Juexiao Zhou[1], Yunfei Li[1], Liang Fang[1,3], Yuhui Hu[1] & Wei Chen[1,3,**]

## Abstract

Alternative polyadenylation (APA) is a major layer of gene regulation. However, it has recently been argued that most APA represents molecular noise. To clarify their functional relevance and evolution, we quantified allele-specific APA patterns in multiple tissues from an F1 hybrid mouse. We found a clearly negative correlation between gene expression and APA diversity for the 2,866 genes (24.9%) with a dominant polyadenylation site (PAS) usage above or equal to 90%, suggesting that their other PASs represent molecular errors. Among the remaining genes with multiple PASs, 3,971 genes (34.5%) express two or more isoforms with potentially functional importance. Interestingly, the genes with potentially functional minor PASs specific to neuronal tissues often express two APA isoforms with distinct subcellular localizations. Furthermore, our analysis of cis-APA divergence shows its pattern across tissues is distinct from that of gene expression. Finally, we demonstrate that the relative usage of alternative PASs is not only affected by their cis-regulatory elements, but also by potential coupling between transcriptional and APA regulation as well as competition kinetics between alternative sites.

**Keywords** alternative polyadenylation; error hypothesis; gene regulation; neutral theory of molecular evolution; regulatory evolution
**Subject Categories** RNA Biology
**Mol Syst Biol. (2020) 16:** e9367

## Introduction

When the transcription of a precursor mRNA reaches its end, it usually undergoes two coupled maturation processes: 3′ end cleavage and polyadenylation (Richard & Manley, 2009; Tian & Manley, 2016). In eukaryotic organisms, transcripts derived from the same gene locus could end at different positions, resulting in distinct isoforms. The phenomenon, termed as alternative polyadenylation (APA), affects more than 70% of protein-coding genes in mammals (Derti et al, 2012; Hoque et al, 2013). APA can lead to the production of alternative protein isoforms, if it involves the splicing of an alternative last exon or the usage of a polyadenylation site (PAS) upstream of the distal stop codon. More often, if alternative PASs are limited to the 3′UTR, it results in transcripts with identical protein-coding sequences, but 3′UTRs of different lengths. These APA isoforms may contain different sets of cis-regulatory elements in their 3′UTRs and therefore may differ in mRNA stability (Chen & Bin, 1995; Barreau et al, 2005; Bartel, 2009; Jonas & Izaurralde, 2015), translation efficiency (De Moor et al, 2005; Lau et al, 2010), intracellular localization (Ephrussi et al, 1991; Bertrand et al, 1998; An et al, 2008; Niedner et al, 2014), or even in the localization and function of the encoded proteins (Berkovits & Mayr, 2015). APA often shows tissue-, cell type-, or developmental stage-specific patterns. In general, distal PAS usage appears to be favored in differentiated cells, whereas proximal PAS usage is more frequently observed in proliferating, undifferentiated, or cancerous cells (Ji & Tian, 2009; Mayr & Bartel, 2009).

Polyadenylation is regulated by various cis- and trans-regulatory elements. In vertebrates, the canonical hexamer AAUAAA and its highly similar variants are the most prominent cis-regulatory elements and defined as polyadenylation signal (Proudfoot & Brownlee, 1976; Proudfoot, 2011). Located 15–30 nucleotides

---

1 Department of Biology, Southern University of Science and Technology, Shenzhen, China
2 Laboratory of RNA Biochemistry, Institute of Chemistry and Biochemistry, Freie Universität Berlin, Berlin, Germany
3 Academy for Advanced Interdisciplinary Studies, Southern University of Science and Technology, Shenzhen, China
*Corresponding author. Tel: +86 181 24086264; E-mail: bernhard@sustech.edu.cn
**Corresponding author. Tel: +86 755 88018449; E-mail: chenw@sustech.edu.cn
†Present address: Department of Pharmaceutical Chemistry and the Cardiovascular Research Institute, University of California San Francisco, San Francisco, CA, USA
‡Present address: Cancer Science Institute of Singapore, National University of Singapore, Singapore City, Singapore
§These authors contributed equally to this work

upstream of the cleavage site, they are directly recognized by the cleavage and polyadenylation specificity factor (CPSF) subunits CPSF30 and Wdr33 (Shi, 2014). The choice of alternative PASs is achieved by a variety of mechanisms. Global APA changes can be introduced by different expression levels of components of 3′ end complexes, e.g., higher expression of CFI would lead to the preference of distal PASs (Martin *et al*, 2012). Among the *trans*-regulatory factors are also RNA-binding proteins (RBPs) which can enhance or repress the binding of the polyadenylation machinery to their target sites (Erson-Bensan, 2016).

Although the function of different APA isoforms has been demonstrated for many individual genes, the phenotypic relevance of most APA observed in modern high-throughput sequencing experiments remains elusive. In a recent study, Xu and Zhang proposed the "error hypothesis", stating that most genes have only "one optimal polyadenylation site and that APA is caused largely by deleterious polyadenylation errors" (Xu & Zhang, 2018). The authors support this hypothesis by showing that in general, lower gene expression levels correlate with higher APA diversity within and between tissues. In extension of Kimura's neutral theory of molecular evolution (Kimura, 1968, 1983), Zhang also suggests that differences between species in gene regulatory phenotypes like APA are largely non-adaptive, as only a small fraction of these will affect higher-level phenotypes and organismal fitness (Zhang, 2018).

If patterns of APA in mammals generally conform to the error hypothesis (concerning the molecular function of APA) as well as the neutral theory (concerning evolutionary patterns of APA), the following predictions can be made: (i) The large majority of genes should have only one optimal PAS and genes subject to the strongest purifying selection would show no APA. (ii) Among the genes with more than one PAS, genes under the lowest levels of selective constraints should exhibit more APA diversity within and across tissues as well as higher divergence between species. (iii) Compared to genes under stronger purifying selection, these genes should show weaker and less conserved APA regulation. Alternatively, the usage of alternative PASs could be functional within and/or between tissues, and changes in APA regulation between species could be adaptive and result from positive selection rather than random drift.

In this study, we test the predictions made by the error hypothesis and neutral theory, by globally characterizing APA patterns across various tissues in the F1 hybrid of the two mouse inbred strains C57BL/6J and SPRET/EiJ, belonging to the closely related species *Mus musculus* (house mouse) and *Mus spretus* (Algerian mouse), respectively. In F1 hybrids, the RNA transcripts from both parental alleles are subject to the same *trans*-regulatory environments; therefore, the observed differences in allele-specific APA patterns reflect the effects of *cis*-regulatory divergence (Wittkopp *et al*, 2004; Xiao *et al*, 2016). As shown in a previous study by our group, evolutionary divergence of APA between closely related species can be attributed largely to *cis*-regulatory changes (Xiao *et al*, 2016). But the extent and general importance of these changes and their interaction with tissue-dependent *trans*-regulation remain undetermined. This dataset therefore enables us to simultaneously study the tissue dependence and potential function of PAS regulation as well as the evolutionary patterns of *cis*-regulatory APA divergence in mammals.

# Results

## Genes with a single polyadenylation site are less conserved and expressed in fewer tissues than multi-PAS genes

To study APA regulation across different tissues, we quantified the PAS usage in eight organs (cortex, cerebellum, heart, muscle, lung, liver, kidney, and spleen) as well as embryonic stem cells (ESCs) from an F1 hybrid between the C57BL/6J and SPRET/EiJ mouse strains, using the Lexogen 3′ mRNA Rev Kit (Materials and Methods). For each tissue, we sequenced three replicates with an average of about 40 million reads per sample, of which approximately 70% uniquely mapped to at least one of the reference genomes (Materials and Methods, Table EV1). We assigned reads to PASs annotated in the polyA_DB3 database (Wang *et al*, 2018a). To assure the reads were derived from the 3′ end, we only counted those reads whose end was located within 24 nucleotides away from the annotated cleavage site (Materials and Methods). On average, 17.15 million reads per sample could be assigned to one of the annotated PASs (hereafter termed "PAS reads"). Among these, an average of 8.14 million reads could be unambiguously assigned to either the SPRET/EiJ or the C57BL/6J genome (Materials and Methods, Table EV1).

We then counted the number of PAS reads for each PAS and the total number of PAS reads mapped to each protein-coding gene. A gene was considered expressed in a tissue only if it was covered by at least 20 PAS reads in each of the three replicates. In total, 13,369 protein-coding genes were expressed in at least one tissue. Among them, 1,859 (13.90%) genes were single-PAS genes, expressing only one identical PAS across all tissues. The remaining multi-PAS genes expressed more than one PAS in at least one tissue or expressed different PASs in at least two different tissues. For each multi-PAS gene in each tissue where it is expressed, we defined the PAS with the highest usage as the "dominant PAS". The PAS with the maximum average usage across all the expressing tissues was defined as a gene's "major PAS".

As previous studies indicated that single-PAS and multi-PAS genes are two distinct classes (Lianoglou *et al*, 2013; Mayr, 2016), we examined their different features based on our data. Consistent with the previous findings, single-PAS genes have on average shorter 3′UTRs (Fig EV1A) and are expressed in fewer tissues (Fig EV1C; this is also true when controlling for sequencing depth, see Materials and Methods and Fig EV1B and D). Their core PAS regulatory regions ($-50$ to $0$ nt) exhibit lower sequence conservation than those of the major PASs in multi-PAS genes (Fig EV1E). In addition, we examined constraints on protein sequence evolution, measured by the ratio of the rate of non-synonymous substitution to the rate of synonymous substitution (dN/dS ratio), with lower ratios reflecting stronger purifying selection on amino acid sequence (Miyata & Yasunaga, 1980). We found that single-PAS genes tend to have higher dN/dS ratios than multi-PAS genes (Fig EV1F), i.e., their protein sequence evolution is also less constrained. In the following analyses, we focused only on the multi-PAS genes (see Dataset EV1 for detailed gene information).

## Genes with distal major PASs have higher gene expression levels and sequence conservation than those with proximal ones

First, we examined the features of multi-PAS genes with different major PAS locations (Fig 1). The major PASs of most multi-PAS

genes (10,893, 94.64%) are located in the 3′UTR annotated in Refseq, whereas only few genes (617, 5.36%) have their major PAS upstream of the annotated last stop codon. Among the major PASs located in the 3′UTR, 1,117 (10.25%) are the only PAS in the 3′UTR. For the remaining 3′UTR major PASs, 2,316 (21.26%) are in the most proximal position, 3,235 (29.70%) in the most distal position, and 4,225 (38.79%) in the middle. In general, major PASs located in the 3′UTR show higher sequence conservation in their core PAS regulatory regions and higher PAS usage (Fig 2A and B); their host genes express at higher levels and have lower dN/dS ratios than those with the major PAS upstream

of the last stop codon (Fig 2C and D). This indicates that genes with major PASs affecting the coding regions are under relaxed selective constraints on protein sequence as well as polyadenylation regulation during evolution. Among the genes with a 3′UTR major PAS, those with a distal or middle location have significantly lower dN/dS ratios and show higher sequence conservation in the core regulatory regions of their major PASs than those with a proximal location (Fig 2A and D). This is also true if we only consider genes with exactly 3 PASs in the 3′UTR (Materials and Methods): 471 of these genes have the major PAS located in the first (proximal) position (3′UTR(F)), 576 in the middle (3′UTR

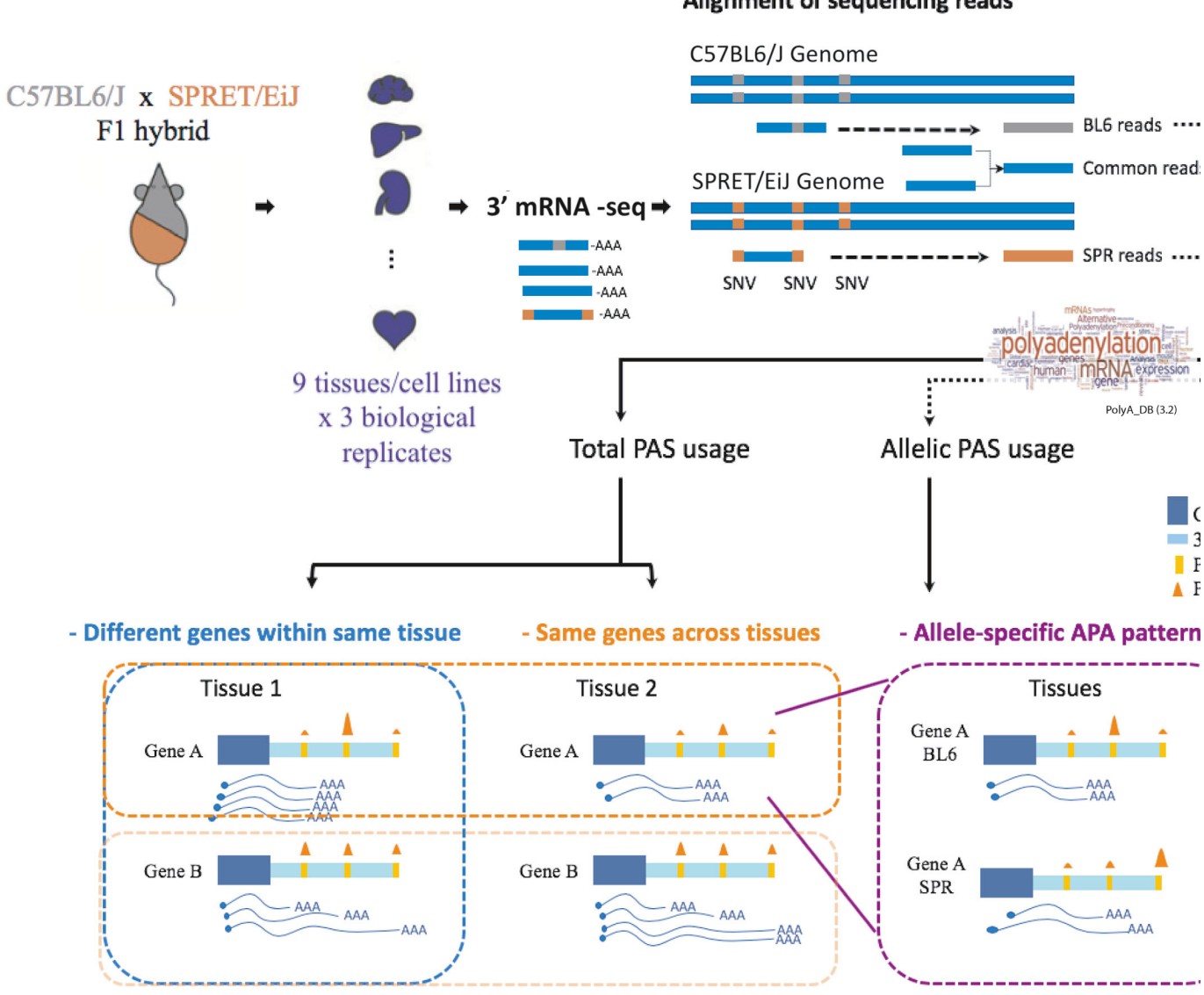

**Figure 1. Experimental and computational analysis scheme of this study.**
Nine tissues/cells from the F1 hybrid of the two mouse inbred strains C57BL/6J and SPRET/EiJ were subject to the 3′mRNA-seq. Sequencing reads were mapped to the reference genomes of the parental strains, respectively, and classified as allele-specific reads or common reads. Only reads uniquely mapped to PASs annotated in the PolyA_DB were used for PAS usage calculation. Total PAS usage (calculated by the sum of allelic and common reads) in different tissues was utilized in within-tissue comparisons and analysis of APA patterns across tissues. The allelic reads were utilized to study allele-specific APA patterns and *cis*-regulatory divergence.

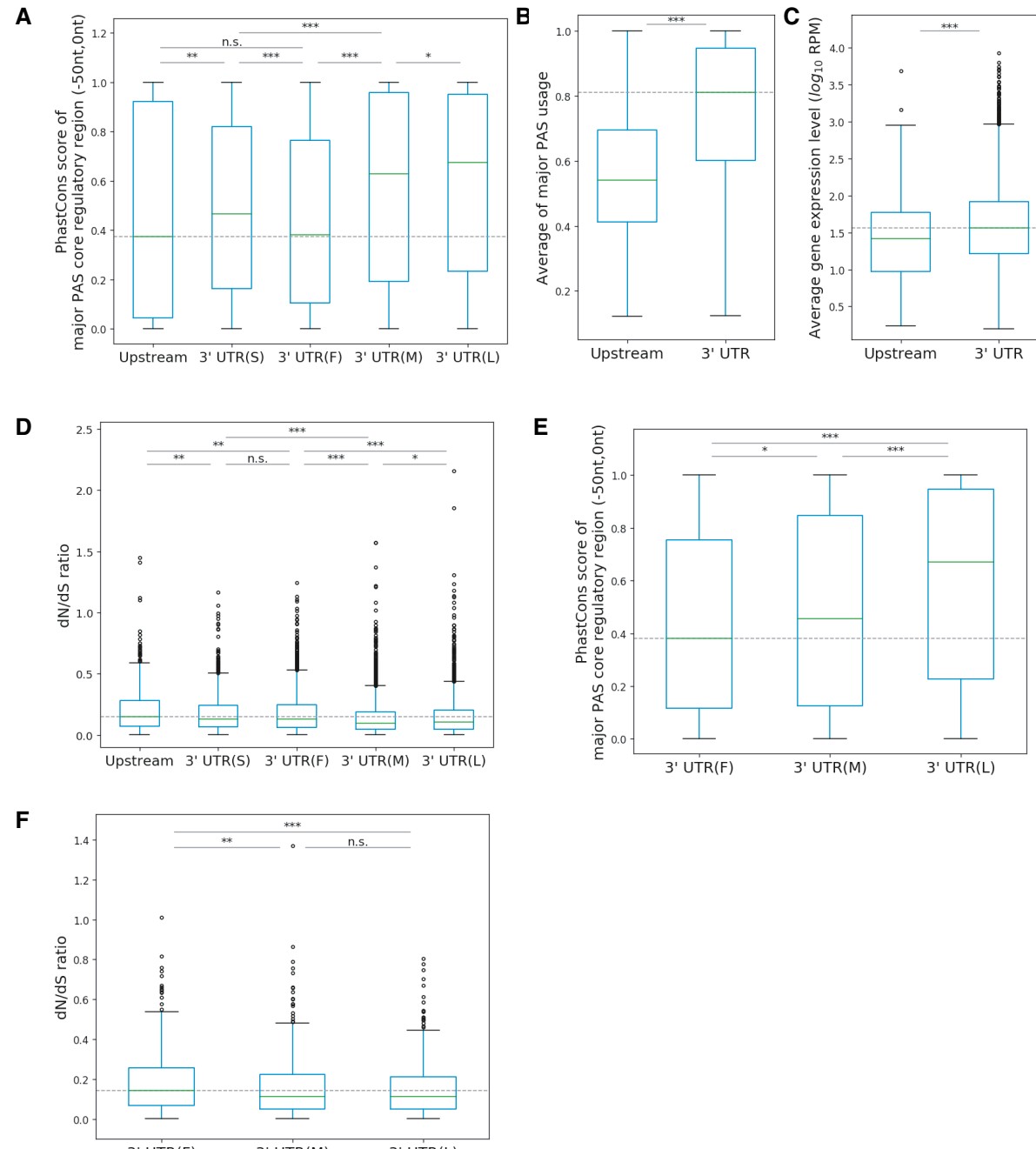

**Figure 2. Genes with distal major PASs have higher gene expression levels and sequence conservation than those with proximal ones.**

A   Sequence conservation comparison of the core PAS regulatory regions between major PASs with different locations. Upstream: upstream of last stop codon, 3′UTR (S): the only PAS in the 3′UTR, 3′UTR(F): the most proximal PAS in the 3′UTR, 3′UTR(M): the middle PAS in the 3′UTR, 3′UTR(L): the most distal PAS in the 3′UTR (Mann–Whitney *U*-test, *P < 0.05; **P < 0.01; ***P < 0.001).

B, C   Genes with major PAS located in 3′UTR have on average higher major PAS usage (B) and gene expression level (C) compared to genes with upstream major PASs (Mann–Whitney *U*-test, ***P < 0.001).

D   dN/dS ratio comparison between genes with different major PAS locations (Mann–Whitney *U*-test, *P < 0.05; **P < 0.01; ***P < 0.001).

E, F   Among genes with exactly three PASs in 3′UTR, genes with major PAS located proximally have lower sequence conservation in the core regulatory regions of the major PASs (E) and are under weaker selective constraint in their coding sequence (F) compared to the other genes (Mann–Whitney *U*-test, *P < 0.05; **P < 0.01; ***P < 0.001).

Data information: The box spans values from the first (Q1) to the third (Q3) quartile, with a horizontal green line indicating the median (Q2). The whiskers extend from the edges of the box to the points representing the largest and smallest observed values within the 1.5 * IQR (interquartile range = Q3 - Q1) from the edges of the box. Outlier values are plotted as dots outside the whiskers.

(M)), and 641 in the last (distal) position (3′UTR(L)), and compared to genes with a proximal major PAS, those with a distal or middle major PAS have higher sequence conservation in the core regulatory regions of the major PAS and exhibit lower dN/dS ratios (Fig 2E and F). These results indicate that major PASs located in the middle or at the distal end of the 3′UTR might be selectively favored, consistent with a previous study showing that 3′UTR(L) PASs are most conserved during evolution (Lee *et al*, 2008). In contrast, genes under relaxed selective constraints might be more likely to evolve a major PAS located proximal to or even upstream of the stop codon.

## Polyadenylation diversity is in general inversely correlated with gene expression but with a complex pattern

For all multi-PAS genes, we then investigated the patterns of polyadenylation diversity within individual tissues. According to the error hypothesis, genes with lower expression levels are expected to have higher polyadenylation diversity than highly expressed genes. To test this hypothesis, we grouped the multi-PAS genes according to their dominant PAS usage or according to the Shannon index of alternative PAS usage, reflecting both the number of different PASs and the evenness of the usage

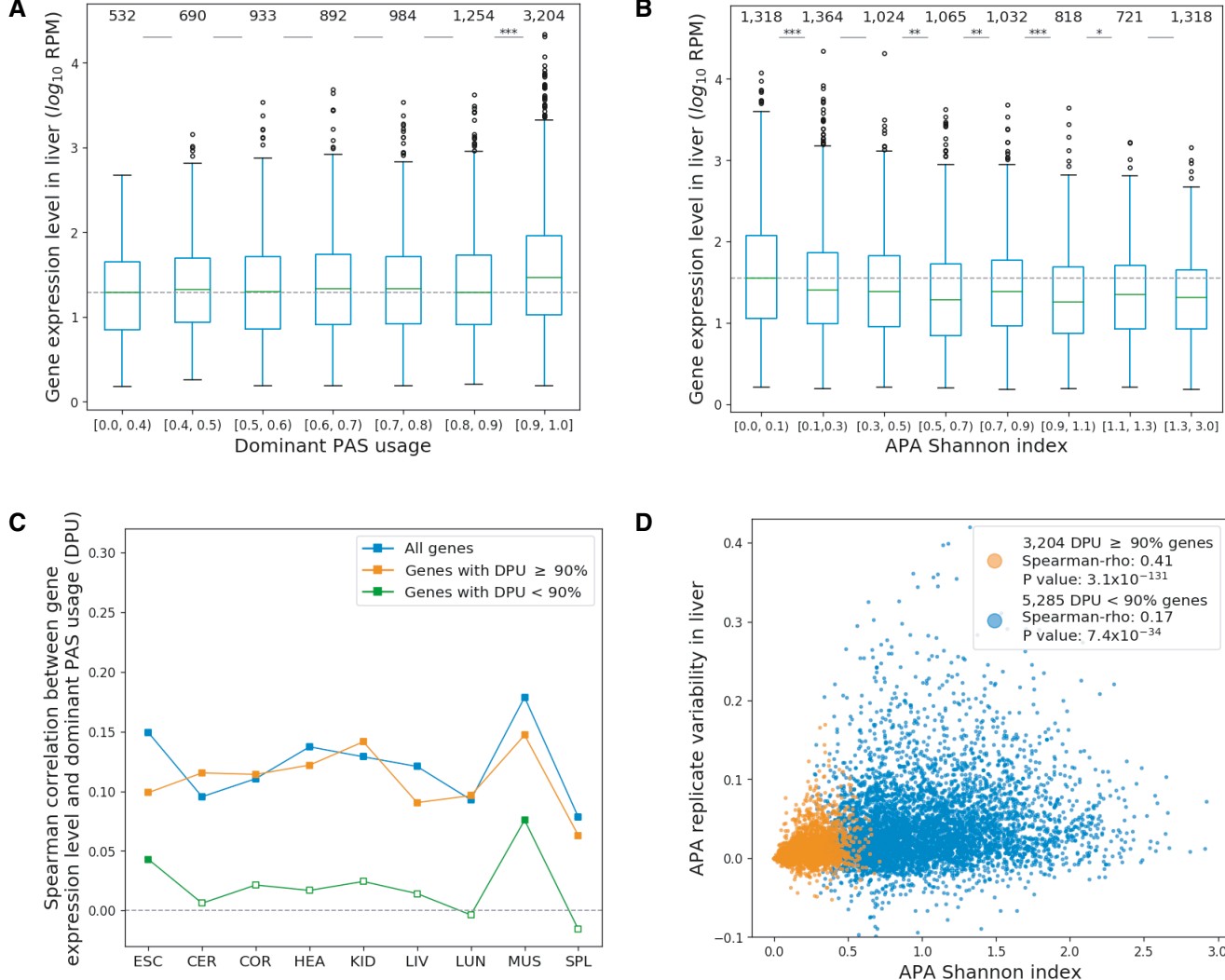

**Figure 3. Multi-PAS genes with low APA diversity have higher gene expression levels.**

A, B Genes with higher dominant PAS usage (DPU) (A) or with lower APA Shannon index (B) express at higher levels (liver shown as example). The number of genes in each group is indicated on top. Mann–Whitney *U*-test significance levels are indicated for neighboring groups: *$P < 0.05$; **$P < 0.01$; ***$P < 0.001$. The boxplots show the Q1 to Q3 quartile values (the box limits), the median (the horizontal green lines), and values within the 1.5 * IQR (the whiskers).

C Spearman correlations between gene expression level and dominant PAS usage (DPU) in each tissue for all genes (blue), genes with DPU equal to or above 90% (orange), and genes with DPU < 90% (green). Genes with DPU < 90% exhibit weaker Spearman correlations between gene expression level and dominant PAS usage in each tissue. An empty square indicates that the *P* value of Spearman's ρ is above 0.01 (ESC: embryonic stem cells, CER: cerebellum, COR: cerebral cortex, HEA: heart, KID: kidney, LIV: liver, LUN: lung, MUS: muscle, SPL: spleen).

D Positive Spearman correlation between APA Shannon index and replicate variability in liver.

distribution across these sites (see Materials and Methods), for each tissue separately. We then compared the distribution of mRNA expression levels between the different gene groups for each tissue. As shown in Figs 3A and B, EV2 and EV3, by and large, multi-PAS genes with higher dominant PAS usage or lower Shannon index expressed at higher levels. This difference is most pronounced between genes with a dominant PAS usage above or equal to 90% and those below that level. Within the latter group, the positive correlation between gene expression and dominant PAS usage is negligible (Fig 3C). This pattern is largely independent of the dominant PAS location within the 3′UTR (Fig EV4A–C). In contrast to those in the 3′UTR, usage of dominant PASs affecting the coding region is not correlated with gene expression level in most tissues.

Two different factors could cause higher APA diversity within a given tissue: 1. higher variability in APA between cells due to increased noise in the polyadenylation process; and 2. consistent high PAS usage diversity across cells in a given tissue. While higher APA noise could be reflected in increased variability also between biological replicates, this would not be predicted in the case of consistent and tightly regulated APA diversity. As expected in the former case, we observed a positive correlation between a gene's APA Shannon index and its adjusted APA variability across three biological replicates in all tissues, in particular for those genes with a dominant PAS usage equal to or above 90% (Materials and Methods, Figs 3D and EV4D), further supporting the error hypothesis for this group of genes.

If the increased polyadenylation diversity in lowly expressed multi-PAS genes is due to error caused by weaker *cis*-regulation in genes under relaxed selective constraints, we would furthermore expect a positive correlation between APA diversity and dN/dS ratios. But this was not the case (Table EV2). This indicates that selective constraints on APA accuracy are largely independent of those on protein sequence during evolution.

## A substantial fraction of genes expresses more than one functional PAS

Although for many genes higher APA diversity is correlated with lower expression levels and higher variability between replicates, and therefore might be attributed to molecular error, some genes might still use two functional PASs in the same tissue. Hypothetically, for these genes, there should be at least one minor PAS with the usage similar to that of the dominant PAS. In each of the 9 tissues, between 14.7 and 16.2% of multi-PAS genes have a rank 2 PAS, i.e., the minor PAS with the second highest usage, with a usage difference of < 20% in comparison with the dominant PAS. In total, 3,778 genes fulfill this criterion in at least one tissue and are therefore considered as potentially using at least two functional PASs (F2 genes). As controls, we considered the 2,866 multi-PAS genes with ≥ 90% dominant PAS usage in all expressing tissues to have only one functional PAS under tight regulation (F1 genes), whereas for the remaining 4,866 genes, the functional relevance of their alternative PASs is not clear (FU genes; Fig EV5). We then examined different features and compared F2 genes to the other two groups.

As shown above, on average, we observed a positive correlation between a gene's APA diversity and its APA variability across

biological replicates, attributing the increased diversity to stochastic noise. However, if we performed the same analysis for the three groups separately, the same positive correlation is only observed for F1 genes and FU genes. In contrast, there is no significant correlation between APA diversity and APA variability for F2 genes in most tissues (Fig 4A), indicating that in genes with at least two functional PASs, APA diversity is under tight regulation.

Next, we focused on the 8,950 genes with both dominant and rank 2 PAS located in the 3′UTR, including 3,036 (80.4%) F2 genes, 2,059 (71.8%) F1 genes, and 3,855 (79.2%) FU genes. It is well known that the 3′UTR often contains *cis*-elements regulating mRNA stability, translation, and localization. If a gene uses two different PASs for producing isoforms with distinct, non-redundant functions in the same tissue, the two mRNA isoforms would harbor different *cis*-regulatory elements. Therefore, if the rank 2 PAS is located downstream of the dominant PAS, the region between the two PASs would be predicted to contain additional *cis*-elements important for the specific regulation of the minor isoform. As expected, the sequence conservation of this region is significantly higher in F2 genes than that in the other two groups (Fig 4B). The hypothesis that the alternative isoforms might be under different post-transcriptional regulation is further supported by the observation that a high density of microRNA target sites was found in this region for F2 genes (Fig 4C).

Because previous studies suggested that microRNA target sites in the 3′UTR are enriched immediately upstream of the PAS (Hoffman *et al*, 2016; Tian & Manley, 2016), we scanned the 300 nt upstream of rank 2 PASs among the three groups. As shown in Fig 4D and E, microRNA target site density and average sequence conservation in the 300 nt upstream of the rank 2 PASs are significantly higher in the F2 group than in the other two gene groups, regardless of the location of the rank 2 PAS (upstream or downstream of the dominant PAS). These findings further support that F2 genes contain functional minor PASs.

It has been known for individual neuronal genes that their mRNA isoforms with distinct 3′UTR could be localized in different subcellular compartments, e.g., somata versus synaptic neuropil (axons and dendrites). To check whether the two isoforms of F2 genes could be differentially localized in neurons, we utilized a genome-wide mRNA localization dataset derived from rat hippocampus (Tushev *et al*, 2018). In total, 6,154 orthologous mouse multi-PAS genes could be found in that dataset with reliable subcellular localization pattern. Whereas only about 10.8% of F1 genes and FU genes have different localization patterns between their mRNA isoforms, 302 (14.2%) of genes with minor functional PASs and 38 (19.7%) of genes with brain-specific minor functional PASs (see Materials and Methods) have isoform-specific localization patterns (Fig 4F). This indicates that the alternative isoforms of many genes with minor functional PASs harbor *cis*-regulatory elements which can regulate mRNA localization.

## APA change across tissues is in general correlated with APA diversity within tissues

We then investigated patterns of APA change across tissues, their relationship with gene expression levels, and APA diversity within tissues. Under the error hypothesis, we would expect that higher changes across tissues reflect noise rather than tissue-dependent

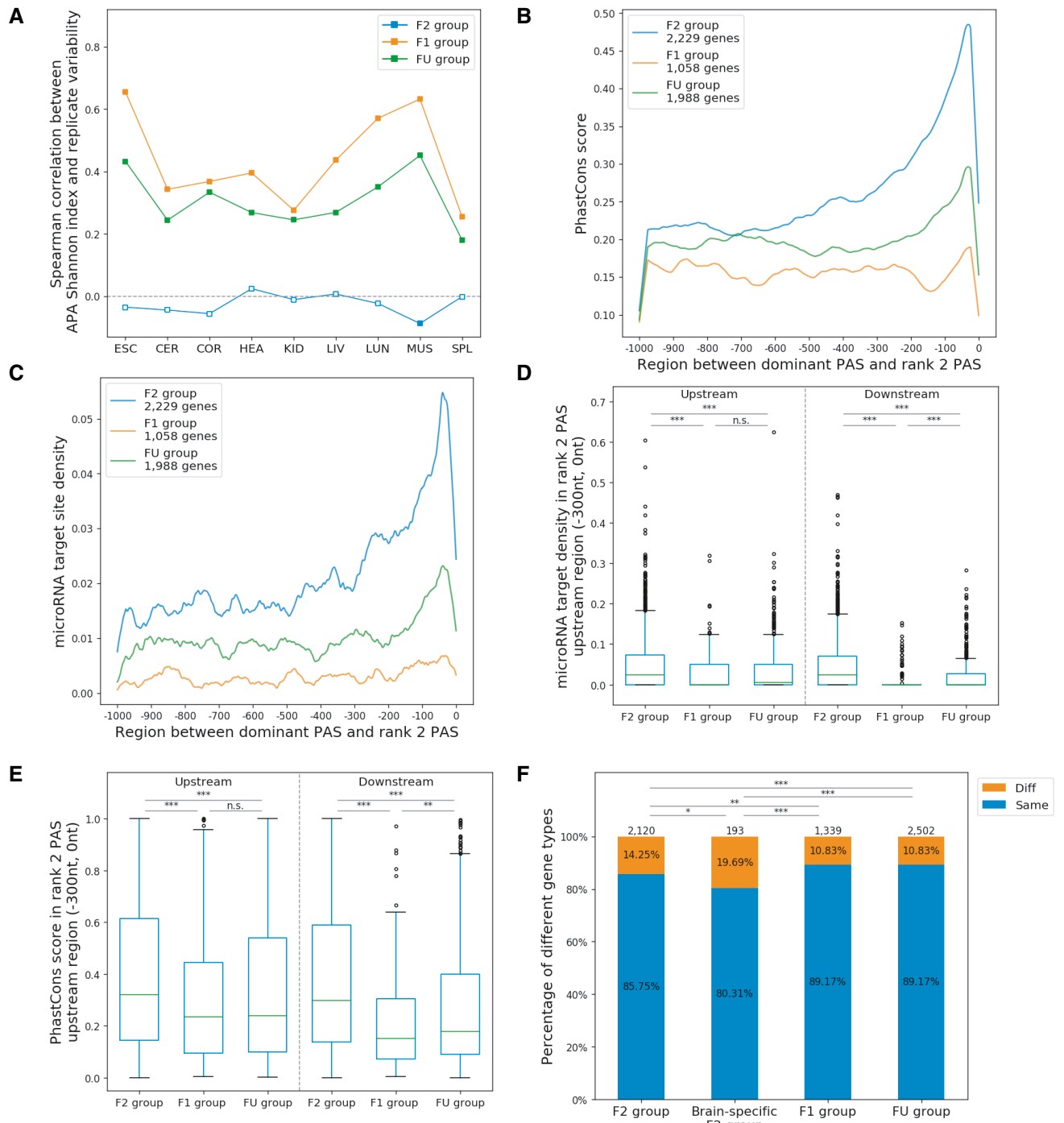

**Figure 4.  Genes with two or more functional PASs versus genes with only one functional PAS.**

A       Spearman correlation between APA Shannon index and APA variability between replicates for each gene group in different tissues. An empty square indicates *P* value of the correlation is larger than 0.01.

B, C    PhastCons score (B) and microRNA target site density (C) in the region between the dominant PAS and the rank 2 PAS of a gene, where the rank 2 PAS is located downstream of the dominant PAS.

D, E    Comparisons of microRNA target site density (D) and PhastCons scores (E) in 300-nt PAS upstream regions among different gene groups. Upstream: rank 2 PAS located upstream of the dominant PAS. Downstream: rank 2 PAS located downstream of the dominant PAS (Mann–Whitney *U*-test, **$P < 0.01$; ***$P < 0.001$). The boxplots show the Q1 to Q3 quartile values (the box limits), the median (the horizontal green lines), and values within the 1.5 * IQR (the whiskers).

F       Fractions of orthologous genes in different groups exhibiting different subcellular localizations of mRNA isoforms in rat neurons. Same: isoforms enriched in same cellular location. Diff: isoforms enriched in different cellular locations, i.e., one isoform in neuropil and another one in soma (Fisher exact test: *$P < 0.05$; **$P < 0.01$; ***$P < 0.001$).

regulation and therefore affect genes with lower expression levels. To test this hypothesis, we assigned a "switch score" for each gene by calculating the maximum difference of PAS usage between expressing tissues. We found that among 6,638 multi-PAS genes with a switch score below 10%, genes with higher switch scores tend to have lower average expression levels (Spearman's ρ: $-0.18$, *P* value: $3.9 \times 10^{-32}$), whereas no negative correlation could be observed for 4,399 genes with a switch score higher than 10% (Spearman's ρ: 0.03, *P* value: 0.02).

Next, we addressed the question whether the magnitude of APA differences between tissues is related to the APA diversity within tissues and the variability between biological replicates of the same tissue. As expected under the error hypothesis, we found that on average, genes with larger differences between tissues also have a higher within-tissue Shannon index and higher APA variability across biological replicates (Figs 5A, and EV6A and B). Again, we found the correlation is more substantial for genes with a switch score below 10%.

### Patterns of APA change across tissues suggest mechanistic coupling between gene expression and polyadenylation accuracy

Under the neutral scenario (larger APA differences between tissues are found in genes under relaxed selective constraints), we would also expect a positive correlation between APA difference across tissues (switch score) and dN/dS ratios. Similar as for within-tissue APA diversity (Table EV2), this was not the case (Spearman's ρ: $-0.03$, *P* value: $6.4 \times 10^{-4}$). This indicates that the relationship between low average expression levels and larger APA differences between tissues cannot be explained by a relaxation of purifying selection simultaneously acting on protein-coding sequence and APA regulation.

Instead, such correlation might be at least partially mediated by a direct mechanistic relationship between transcription/mRNA abundance and polyadenylation accuracy. As shown in Fig 5B, we found predominantly positive correlations between the expression level of a gene and its major PAS usage across tissues (the tissue with higher gene expression tended to show higher major PAS usage). This further corroborates that for most genes, higher expression levels lead to more accurate polyadenylation, if higher dominant PAS usage reflects higher polyadenylation accuracy. Interestingly, such correlation between gene expression level and major PAS usage is stronger for genes where the major PAS is

located distal or middle in the 3'UTR than for genes with proximal major PASs (Fig 5C). To further investigate the effects of PAS location on PAS usage across tissues, we again focused on genes with exactly 3 PASs in their 3'UTRs. We found that generally the expression levels of a gene across tissues positively correlate with increased usage of the major PAS (as observed for all multi-PAS genes). However, if the major PAS is in the middle, gene expression level is also positively correlated with increased usage of the last (most distal) minor PAS (Fig 5D), notably, even with higher magnitude than for the major PAS usage. This indicates that mechanistic or functional coupling between transcriptional and APA regulation might generally lead to increased usage of distal PASs in highly expressed genes.

### Some functional minor PASs are regulated in a tissue-dependent manner

Although different PAS usage between tissues in many genes may reflect non-functional noise, some of the larger APA differences observed between tissues could be caused by functional tissue-dependent regulation. To search for the latter, we defined a gene as "switched gene" if there is at least one PAS showing a usage difference above or equal to 50% in any pairwise tissue comparison. Whereas there is a significant positive correlation between APA difference and APA variability for the genes with smaller PAS usage differences between tissues, this is not the case for switched genes, which generally exhibit lower APA variability between replicates (Fig 5E), indicating that the tissue-dependent APA differences are under stringent regulation. In total, we found 724 switched genes in all pairwise comparisons between tissues, including 528 F2 genes, 3 F1 genes, and 193 FU genes according to the previous classification (Fig EV5). These results indicate that APA patterns in F1 genes generally do not have any drastic tissue-dependent changes. Instead, F1 genes use only one functional PAS across all tissues. In contrast, among F2 and FU genes with both the major and the 2nd most used PAS located in the 3'UTR (2,649 genes in F2 group and 3,240 genes in FU group), the switched genes (273 genes in F2 group and 70 genes in FU group) have much higher sequence conservation and microRNA target site density in the regions between the two PASs compared to the non-switched genes (2,376 genes in F2 group and 3,170 genes in FU group) in both groups (Fig EV6C and D), indicating that tissue-dependent switch between the two most used PASs might be functionally relevant in these genes.

▶

---

**Figure 5. Genes with different APA patterns across tissues.**

A Positive correlation between a gene's median APA Shannon index across tissues and its APA switch score.

B Spearman correlations between differences in major PAS usage and in gene expression level for each tissue pair comparison (*P* value < $4 \times 10^{-10}$ in all comparisons).

C Spearman correlation between the expression level of a gene and its major PAS usage for genes with different major PAS location. *P* values were obtained from binomial tests to examine whether the number of genes with positive correlations is larger than the random expectation of 50%: ***$P$ < 0.001. Upstream: upstream of last stop codon, 3'UTR(S): the only PAS in the 3'UTR, 3'UTR(F): the most proximal PAS in the 3'UTR, 3'UTR(M): the middle PAS in the 3'UTR, 3'UTR(L): the most distal PAS in the 3'UTR. The boxplot shows the Q1 to Q3 quartile values (the box limits), the median (the horizontal green line), and values within the 1.5 * IQR (the whiskers).

D Spearman correlation between the expression level of a gene and its first, middle, and last PAS usage for genes with exactly 3 PASs in 3'UTR and major PAS in 3'UTR (M). *P* values are from binomial tests, **$P$ < 0.01; ***$P$ < 0.001. The boxplot shows the Q1 to Q3 quartile values (the box limits), the median (the horizontal green line), and values within the 1.5 * IQR (the whiskers).

E Spearman correlation between tissue APA differences and APA variability between replicates in each tissue pair.

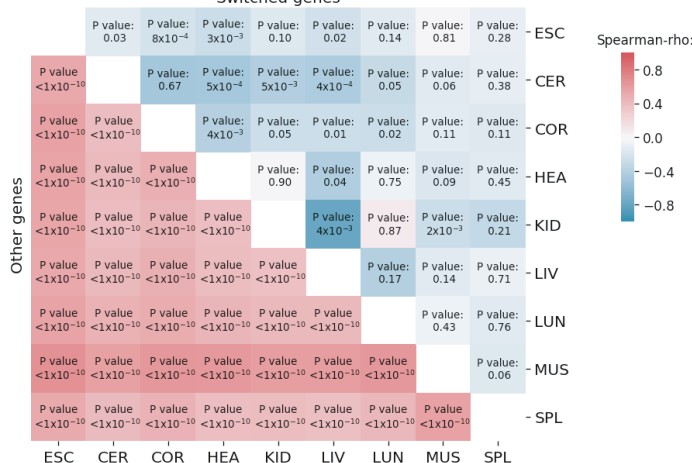

**Figure 5.**

## A subset of 2,218 multi-PAS genes are high-confidence candidate genes with functional APA

As shown above, a large fraction of F2 genes and switch genes appears to have two or more functional PASs and the two groups overlap to a significant degree. Therefore, we defined the union of F2 genes and switched FU genes (comprising 3,971 genes) as "APA functional type" genes. For F1 genes in contrast, minor PAS usage most likely reflects molecular error.

We utilized the cleavage and polyadenylation site information for humans and mouse deposited in the polyA DB database (Wang *et al*, 2018b) and compared the F1 and "APA functional type" genes for their PAS conservation. We found that genes with two or more potentially functional PASs indeed had larger numbers of conserved minor PASs than F1 genes: Whereas 2,443 (61.52%) "APA functional type" genes had two or more conserved PASs with humans, this was the case only for 853 (29.77%) F1 genes (Fig EV7A). To compile a high-confidence set of candidate genes with functional APA more stringently, we selected all genes from the "APA functional type" group which either showed a positive correlation across tissues between gene expression level and usage of a potentially functional minor PAS (secondary PAS) or have an expression level higher than the median of all expressed genes in the tissue with the highest usage of the secondary PAS. In total, 2,218 genes fell into this category. Among these, 1,571 (70.83%) contain two or more conserved PASs with *Homo sapiens* and might therefore be of functional relevance also in humans.

Next, we were interested in finding candidates for functional APA affecting the coding region. As shown in our results, in many cases the usage of PASs upstream of the last stop codon can be attributed to molecular error. This might be different if APA is coupled with alternative splicing of the last exon, as recent evidence shows the functional importance of different protein-coding isoforms produced in this way and their differential localization in neurons (Taliaferro *et al*, 2016). An additional indication of functional APA is evolutionary conservation of the PASs involved. Among the genes with both the dominant and the second PASs conserved between humans and mouse, we found in total 171 genes with APA involving alternative last exon splicing. Among the eight F1 genes in this group, only one contains conserved C-terminal peptides specifically encoded by the minor isoforms (Fig EV7B). In contrast, 102 out of 121 genes with potentially functional APA exhibit conserved amino acid sequences encoded specifically by the minor isoforms between humans and mouse (76 out of the 102 genes are high-confidence candidates; see Dataset EV2 for the detailed information). For two of these genes, *Lmna* and *Cdc42*, both isoforms have been shown to exhibit distinct functions in previous studies. The *Lmna* gene encodes the protein lamin, which functions in nuclear mechanics and genome regulation (Gruenbaum & Foisner, 2015). As shown in Fig EV7C, *Lmna* gene expressed two APA isoforms in all the eight tissues as well as the ES cells. Compared to the long isoform lamin-A, the shorter isoform (lamin-C) lacks the amino acid residues encoded by exons 11 and 12. It does not contain the C-terminal isoprenylation motif (CaaX motif) and therefore cannot be lipidated, in contrast to the longer lamin-A isoform (Fisher *et al*, 1986). Importantly, the lamin-C-only-expressing mice were reported to have reduced mitochondrial activity and energy expenditure as well as increased fat mass and lifespan, demonstrating the functional importance of both APA isoforms (Lopez-Mejia *et al*, 2014). The *Cdc42* gene encodes a small GTPase of the Rho family which regulates cellular morphology and neurite outgrowth in the brain (Kang *et al*, 2008). It produces a brain-specific isoform with a 3′-terminal exon 6 (Cdc42E6) and a ubiquitous isoform with a 3′-terminal exon 7 (Cdc42E7; Fig EV7D). It has been shown that the two isoforms exhibit distinct subcellular localization patterns in neurons determined by their different 3′UTRs (Mattioli *et al*, 2019). Even more, Cdc42E6 can promote spine induction more robustly than Cdc42E7 and is responsible for activity-driven synapse remodeling (Kang *et al*, 2008).

### *Cis*-regulatory APA divergence between *Mus musculus* and *Mus spretus* predominantly affects genes under relaxed selective constraints

We then investigated allele-specific differences in PAS usage across tissues (see Materials and Methods), reflecting *cis*-regulatory divergence between the two mouse strains. Here, we defined a gene as divergent if the difference in allelic PAS usage of the dominant PAS (choosing the C57BL/6J allele arbitrarily as the reference) is above 10% and the Benjamini–Hochberg-adjusted $P$ value is < 0.05 (Materials and Methods). In total, there are between 5,154 and 7,009 genes expressed in individual tissues with sufficient numbers of allele-specific PAS reads to determine the allelic divergence (Materials and Methods), among which between 13.6% (ESC) and 28.5% (spleen) are divergent between the two alleles.

---

**Figure 6.  Divergent genes are under relaxed selective constraints.**

A   APA divergent genes have higher dN/dS ratios compared to conserved genes in each tissue (Mann–Whitney *U*-tests, **$P$ < 0.01; ***$P$ < 0.001). The boxplot shows the Q1 to Q3 quartile values (the box limits), the median (the horizontal green line), and values within the 1.5 * IQR (the whiskers).

B   Divergent genes have higher APA variability between replicates compared to conserved genes in each tissue (Mann–Whitney *U*-tests, ***$P$ < 0.001). The boxplot shows the Q1 to Q3 quartile values (the box limits), the median (the horizontal green line), and values within the 1.5 * IQR (the whiskers).

C   Spearman correlation between allelic difference in dominant PAS usage and total gene expression level (shown for liver as an example).

D   Frequency of sequence variants in major PAS flanking region (-200 nt, 100 nt) for all-divergent genes, some-divergent genes, and conserved genes. The numbers of genes in each group are indicated in the legend.

E   Scaling law, major PASs with moderate usage have larger allelic PAS usage difference (Mann–Whitney *U*-tests, *$P$ < 0.05; ***$P$ < 0.001). The boxplot shows the Q1 to Q3 quartile values (the box limits), the median (the horizontal green line), and values within the 1.5 * IQR (the whiskers).

F   Similar SNV densities in PAS flanking regions (−200 nt, 100 nt) of major PASs between genes with different major PAS usage (Mann–Whitney *U*-test, **$P$ < 0.01). The violin plot shows the median (the white dots), Q1 to Q3 quartile values (the thick black lines), 1.5 × IQR (the thin black lines), and kernel density distribution with the range of the observed data.

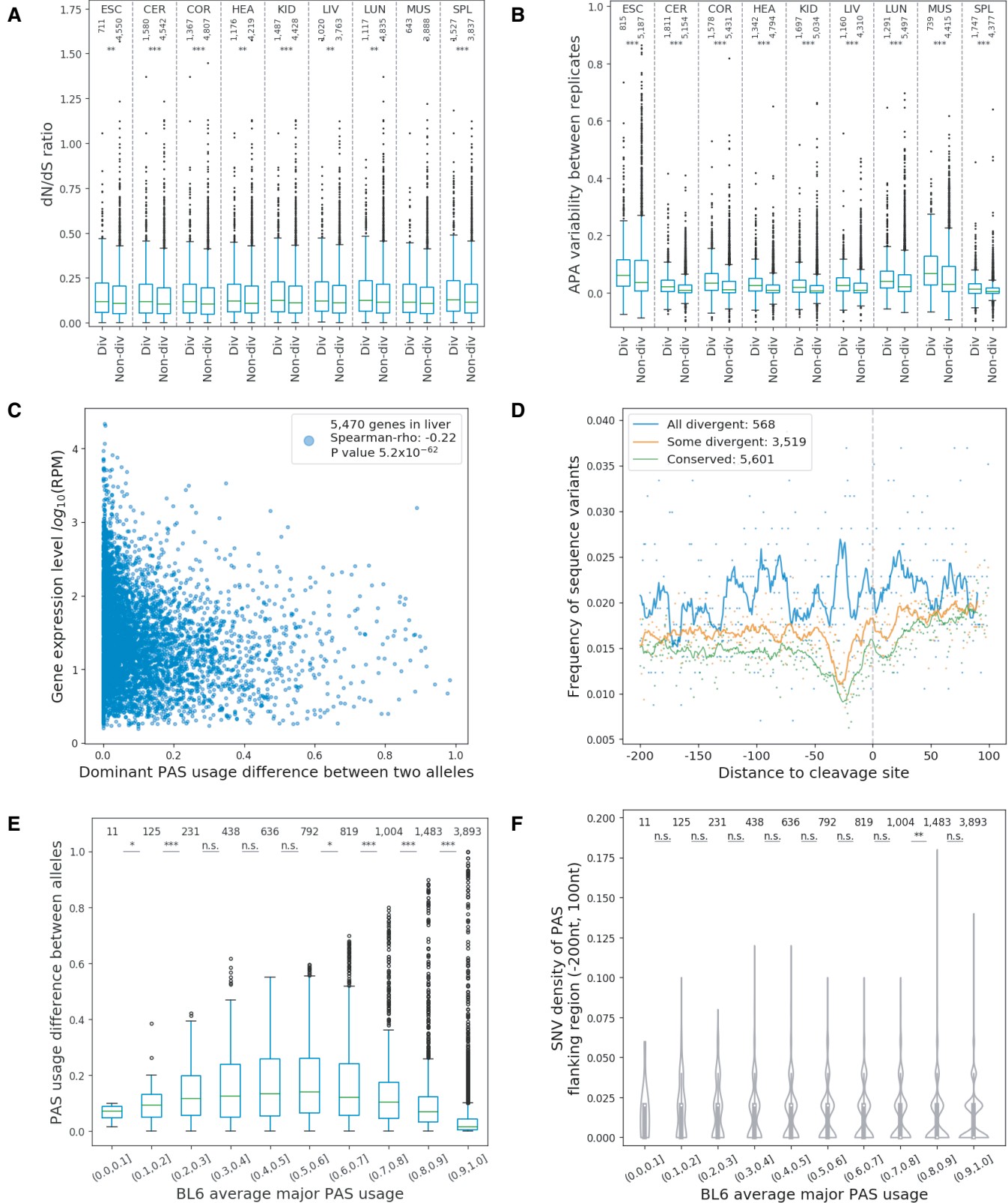

**Figure 6.**

As predicted by the neutral hypothesis, divergent genes have in general higher dN/dS ratios compared to non-divergent genes (Fig 6A), indicating genes with divergent APA patterns have been under relaxed selective constraints. Additionally, divergent genes showed higher APA variability between replicates than non-divergent genes (Fig 6B), again suggesting that allelic APA divergence more likely represents molecular noise.

Furthermore, we found that genes with lower expression levels tended to show APA differences of larger magnitude between the two strains (Fig 6C, shown for liver as example). This could be explained by two different, but not mutually exclusive scenarios: (i) Genes with low expression levels are under reduced purifying selection and accumulate *cis*-regulatory mutations affecting APA faster than more conserved genes. (ii) Genes with lower expression levels frequently have lower levels of major PAS usage (higher APA diversity) than highly expressed genes (as shown above). In such a gene with two or more PASs with similar usage rates, a single *cis*-regulatory mutation could lead to divergence of greater magnitude than in genes with high major PAS usage, as a small change in strength of one PAS would have a larger impact on PAS choice probabilities in a competing pair of similar strengths than in a pair with a large difference in strengths. This kind of scaling law was recently observed in alternative splicing and mathematically described based on the kinetics of competition between splice sites (Baeza-Centurion *et al*, 2019).

To examine whether one or both of these scenarios can explain our findings, we divided all genes into three classes: non-divergent genes (no significant APA divergence in any expressing tissues), some-divergent genes (APA divergent in some expressing tissues, but not all), and all-divergent genes (APA divergent in all expressing tissues). To estimate whether divergent genes might have accumulated *cis*-regulatory mutations at a faster rate, we compared the density of allelic sequence variants in the PAS flanking regions (-200 nt, 100 nt) among the three groups. We found, as expected, that all-divergent genes have the highest variant densities and non-divergent genes the lowest (Fig 6D). Then, to examine a possible scaling law, we asked whether in genes with divergent APA, the major PASs with moderate usages had larger differences between the two alleles than those with higher usages (choosing the C57BL/6J allele arbitrarily as the reference). This was indeed the case (Fig 6E). Importantly, genes with different major PAS usage (Fig 6F) have similar SNV densities in their major PAS flanking regions ($-200$ nt, 100 nt). These results therefore indicate that both *cis*-regulatory sequence variants and a scaling law affect APA divergence.

### Tissue-specific regulation of mRNA expression levels is more conserved and complex than APA regulation

Different layers of gene regulation might exhibit different evolutionary patterns due to different selection pressure and/or differences in the underlying molecular mechanisms. Using our dataset, we compared allele differences in APA with the allelic difference in mRNA levels. As shown in Fig 7A and B, if based on APA patterns (see Materials and Methods), samples are clustered first by allele, then by tissue, whereas if based on mRNA abundance, they are clustered first by tissue, then by allele. It appears therefore that tissue-dependent gene expression patterns are generally more conserved than tissue-dependent APA patterns. This conclusion is further

supported by the observation that APA change across tissues is more strongly correlated with variability between replicates (Fig EV6B) than is the case for gene expression level (Spearman's ρ: $-0.10$, P value: $2.3 \times 10^{-24}$), indicating that noise plays a relatively larger role for tissue differences in APA compared to gene expression. Tissue-specific regulation in gene expression is therefore much more prominent and functionally relevant than that in APA.

We then compared the tissue-dependent divergence between APA and mRNA abundance. Interestingly, as shown in Fig 7C and D, for genes with APA divergence in more than one tissue, there are very few cases where the allele difference shows opposite direction (40 genes, 0.98%), whereas for mRNA abundance, as many as 214 genes (4.89%) show allelic differences in opposite directions (Fisher exact test P value: $7.79 \times 10^{-27}$). This is likely due to differences in the complexity of the underlying regulatory networks, i.e. APA is regulated by a smaller number of *trans*-factors compared to gene expression and/or the expression of APA *trans*-factors varies less than that of transcription factors across tissues (Shi *et al*, 2009; Lambert *et al*, 2018).

## Discussion

APA is a common process in higher eukaryotes which has received increased attention in recent years with regard to its function, regulation, and evolution. Improved high-throughput deep sequencing technology has enabled researchers to identify a growing number of genes undergoing APA, and it is often assumed that the detected variation in PAS usage is biologically relevant. However, cases where functions of different APA isoforms have been clearly demonstrated are still rare compared to the ubiquity of the phenomenon. It has therefore been proposed that most observed APA is noise ("error hypothesis"; Xu & Zhang, 2018) and that APA divergence between species is largely neutral or slightly deleterious ("neutral theory"; Zhang, 2018). Here, we address these issues in an F1 mouse model by investigating APA diversity within each tissue, APA differences between tissues, and *cis*-regulatory divergence between *Mus musculus* (C57BL/6J) and its sister species *Mus spretus* (SPRET/EiJ). We show that the error hypothesis does not predict the differences between single-PAS and multi-PAS genes and that it only applies to a specific subset of multi-PAS genes. Furthermore, we demonstrate that higher APA diversity or divergence can only partially be attributed to a relaxation of selective constraint on lowly expressed genes. Our results indicate that functional coupling between transcriptional and APA regulation (with higher expression levels leading to increased distal PAS usage) and competition between different PASs constitute additional factors with a significant impact on APA variation. Finally, we provide a set of genes with likely functional APA.

Different selective constraints might be reflected in different major PAS locations. We found that genes with the major PAS located upstream of the stop codon have lower expression levels and higher dN/dS ratios than all the other genes with major PAS in the 3′UTR (Fig 2C and D). Whereas for the latter generally gene expression levels are positively correlated with major PAS usage, this is not the case for the former (Fig EV4A–C), indicating that usage of many major PASs affecting the coding region might be deleterious. Indeed, the major PASs located upstream of the stop codon also have lower usage compared to the 3′UTR major PASs.

Among genes with major PAS located in the 3′UTR, genes with major PAS located at a more distal position have lower dN/dS ratios than those proximal to the stop codon. These results together demonstrate that major PAS location is related with selective constraints both on protein-coding sequence and PAS choice

accuracy, with distal UTR PASs being selectively favored over proximal UTR PASs and these in turn over PASs upstream of the stop codon.

The error hypothesis predicts stronger noise in APA regulation and therefore higher APA diversity within and across tissues in

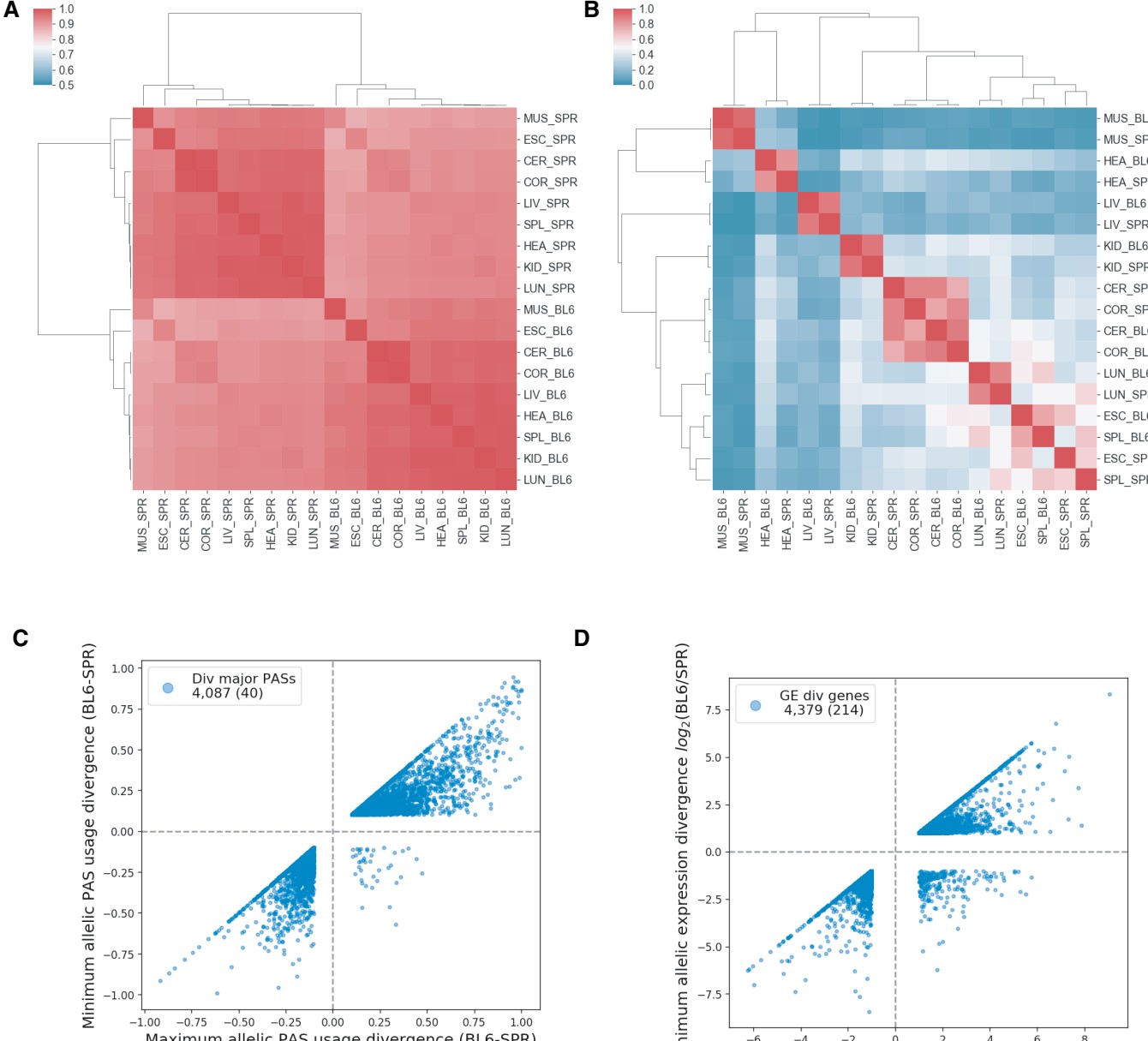

**Figure 7. Evolutionary conservation patterns in APA and gene expression.**

A   2D clustering of Pearson correlation coefficients between all pairwise sample combinations of C57BL/6J (BL6) or SPRET/EiJ (SPR) allelic PAS usages from 5,264 genes (see Materials and Methods).

B   2D clustering of Pearson correlation coefficients between all pairwise combinations of BL6 or SPR allelic gene expression levels from 5,264 genes. All clustering was performed hierarchically using Pearson correlations between samples.

C   Scatterplot of maximum (*x*-axis) and minimum (*y*-axis) allelic PAS usage divergence (BL6 - SPR) across tissues for each gene. The total number of APA divergent genes is indicated in the legend. The number of APA divergent genes with opposite sign between maximum and minimum is in parentheses.

D   Scatterplot of maximum (*x*-axis) and minimum (*y*-axis) allelic gene expression divergence $\log_2$(BL6/SPR) for each gene. The total number of gene expression divergent genes is indicated in the legend. The number of gene expression divergent genes with opposite sign between maximum and minimum is in parentheses.

genes with lower functional impact. In general, these would be genes under relaxed selective constraints. When the relationship between gene expression levels and APA diversity within a tissue or APA changes across tissues is considered, the multi-PAS genes with major PAS located in 3′UTR can largely be separated into two distinct groups: 1. genes with dominant PAS usage above or equal to 90% within each tissue (F1 genes, Fig EV5), which show strong positive correlations between gene expression level and dominant PAS usage and 2. the remaining genes with more diverse APA, for which the negative correlations between gene expression and APA diversity become much weaker and even negligible (Fig 3C). This suggests that for the former group, all the other alternative PASs can be safely regarded as noise, while within the latter group, some genes might contain functional minor PASs (see Discussion below, Fig EV5).

In contrast to gene expression levels, dN/dS ratios, as an alternative measure of selective constraints, however, do not show any correlations with APA diversity/change. At least two possible factors could contribute to the discordance between different measures of selective constraints. One possibility is that some genes with lower average expression levels experience relaxed selective constraints on APA regulation, but not on coding sequence evolution, i.e., purifying selection acts independently on the two. This would be the case, for example, when a different version of the protein has a toxic effect on the cell even in low quantities, whereas low levels of a minor mRNA isoform with a different 3′-UTR length would have little impact on organismic fitness. Alternatively, the regulation of gene expression level and APA accuracy is mechanistically or functionally coupled. There could be three non-mutually exclusive scenarios of such coupling: (i) General *trans*-factors are a limited resource and highly expressed genes compete more successfully for their binding (e.g., through increased recruitment and higher localized concentrations of *trans*-factors), resulting in increased polyadenylation accuracy compared to lowly expressed genes. (ii) Specific *trans*-factors increasing the cleavage and polyadenylation accuracy of their target genes are co-regulated with these genes in a tissue-dependent manner. For example, genes with the major PAS located in the 3′UTR proximal (distal) to the stop codon could be more highly expressed in tissues with higher abundance of *trans*-factors favoring general 3′UTR shortening (lengthening; MacDonald & McMahon, 2010; MacDonald, 2019), or genes containing specific *cis*-regulatory sequences might be co-regulated with the *trans*-factors targeting these *cis*-elements. This might result in increased polyadenylation error of genes in the tissues where the gene and its regulators both are lowly expressed. (iii) Transcriptional regulation is mechanistically coupled with polyadenylation in a way that automatically leads to increased major PAS usage with increased mRNA expression levels. Consistent with such mechanistic coupling, we found that the usage of the most distal PAS (3′UTR (L) PAS) for each gene was strongly correlated with gene expression level across tissues, even for the minor distal PASs (Fig 5C and D), i.e., distal PAS usage of a given gene automatically increases with higher expression levels. Recently, a study in *Drosophila* showed that faster elongation rates might lead to increased usage of distal PASs in the body, but not in brain (Liu *et al*, 2017). Therefore, the mechanistic basis for this phenomenon and the relationships between transcriptional activation, elongation rate, PAS choice, and

mRNA abundance warrant further experimental investigation in mammalian organisms.

In addition, the magnitude of PAS usage differences between tissues or between alleles might depend on the kinetics of competition between different PASs within the same gene. Indeed, we observe that major PASs with intermediate usage (and therefore higher APA diversity) more likely show larger magnitudes of allelic divergence, a scaling law similar to that previously described for alternative splicing (Baeza-Centurion *et al*, 2019). This indicates that the same *cis*-regulatory mutation could cause changes of different magnitude dependent on the relative strengths of other competing PASs.

Taken these two together, our results suggest that (i) the transcription and polyadenylation machineries have co-evolved to ensure that energy and limited resources are preferentially used for the accurate cleavage and polyadenylation of highly transcribed genes; and (ii) the scaling law ensures that the subtle stochastic regulation, due to either *trans*- or *cis*-perturbation, would not cause dramatic usage changes for functionally important PASs with high usage, particularly those with usage higher than 90%. On the other hand, stronger purifying selection would be required to maintain equal usage of two competing functionally important PASs in the same gene. In the future, it will be important to investigate the mechanism(s) which lead(s) to better accuracy of highly expressed genes as well as the role of competition between PASs in shaping the response to regulatory perturbations, e.g., mutations in *cis*-elements and fluctuations in *trans*-factor expression levels.

Finally, we showed that at least 3,778 multi-PAS genes (F2 genes, Fig EV5) express potentially functional minor PASs simultaneously in at least one tissue. Two lines of evidence support their functional relevance: (i) The conservation scores of the 300-nt upstream regions of these minor PASs are similar to those of major PASs, and (ii) the regions between the minor PASs and the major PASs in these genes are more conserved than in other genes and these regions contain higher densities of microRNA binding sites. Functional minor PASs apparently are particularly important in the brain, where the two isoforms are often localized in different subcellular compartments. A significant proportion of genes with two potentially functional PASs in the same tissue also show tissue-dependent regulation. Interestingly, switch-like regulation of different isoforms is also found in 193 FU genes (Fig EV5), and in these genes, the regions between the dominant and the rank 2 PAS have higher conservation scores and microRNA densities than in other FU genes (Fig EV6C and D). Genes with switch-like tissue regulation also are largely consistent in their relative isoform usage between alleles, even if the genes are classified as divergent. There are only 2 genes out of 105 divergent switched genes exhibiting different allelic PAS change directions between the two tissues (Fig EV8). This implies that the influence of tissue-dependent APA regulation exceeds the molecular noise reflected in the divergence within a given tissue. Within the union of switch genes and F2 genes (3,971 genes in total), we define a subset of 2,218 high-confidence candidates for functional APA regulation with high gene expression levels in tissues with significant minor PAS usage. Among these, 1,571 (70.83%) contain two or more conserved PASs with *Homo sapiens* and might therefore be of functional relevance also in humans. These results, together, provide a prioritized set of APA isoforms for future functional studies.

                                      

# Materials and Methods

## RNA extraction from tissues and cultured cells

Tissues including cortex, cerebellum, heart, lung, liver, spleen, kidney, and muscle were obtained from three adult F1 hybrid mice (C57BL/6J x SPRET/EiJ mouse strains). The mice were sacrificed by cervical dislocation, and tissues were dissected and stored in −80°C before RNA extraction. All experiments were carried out in accordance with guidelines for the use of laboratory animals and were approved by the local ethical committee (Pasteur Institute). The embryonic stem cells (ES cells) were cultured in Neurobasal-DMEM F12 (Gibco) with N2B27 (Invitrogen), 2i (Selleck), and LIF (Millipore). Total RNA was extracted from all tissues and cells using TRIzol reagent according to the manufacturer's protocol (Life Technologies). The integrity of purified total RNA was estimated by Agilent Bioanalyzer using RNA Nano Kit (Agilent Technologies) before subsequent experiments. Total RNA with an RNA integrity number (RIN) above 9.0 was used for 3′ mRNA library preparation.

## 3′ mRNA library sequencing

QuantSeq 3′ mRNA-Seq Library Prep Kit REV for Illumina (Lexogen) was used to sequence the 3′ end of RNA with polyA tail. In brief, 500 ng total RNA was taken as input. RNA 3′ end regions were reverse-transcribed using an anchored oligo-dT primer, and second strand synthesis was initiated by random priming. PCR amplification was then performed to obtain an Illumina compatible sequencing library. All libraries were sequenced in paired-end 2 × 151 nt format on an Illumina HiSeq X Ten machine.

## Alignment of sequencing reads

The C57BL/6J reference genome (mm10) was downloaded from Ensembl (https://ensembl.org/info/data/ftp/index.html). The SPRET/EiJ reference genome was created as described previously (Gao et al, 2015). The reference set of PASs was obtained from PolyA_DB3 (http://exon.umdnj.edu/polya_db/), which provides precise cleavage positions determined by 3′ READS method (Zheng et al, 2016). We converted the PAS reference in mm9 to mm10 coordinates using g2gtools (version 0.1.29). Genes with overlapping annotated PASs were removed from further analysis.

For the 3′ mRNA-seq data, the 3′ adaptor sequence 5′-AGATCG GAAGAGCACACGTCTGAACTCCAGTCAC-3′ was first removed from forward reads and 5′-AAAAAAAAAAAAAAAAAAAAGATCG GAAGAGCG TCGTGTAGGG-3′ from reverse reads by Cutadapt (version 1.18). The paired reads with each read longer than 15 nt were mapped to the mm10 and SPRET/EIJ genomes separately by HISAT2 (version 2.0.1) with parameters---no-softclip---no-discordant -x. Reads mapped to at least one of the reference genomes were used. In these mapped reads, if a read could map uniquely to one genome with a shorter edit distance than to the other, it was assigned as unambiguously mapped to the specific allele with the shorter edit distance. If a uniquely mapped read had the same edit distance to both parental genomes, it was assigned as common read. To unify the coordinates, reads assigned to the SPRET/EIJ allele were converted to mm10 coordinates by g2gtools. We defined all the uniquely mapped forward reads with their 5′ end within 24 nt away from the annotated cleavage positions as PAS reads.

## PAS usage quantification

Genes with a sum of < 20 PAS reads in each sample were defined as not expressed and were not considered for further analysis. A PAS was considered as used if at least five reads were assigned to the PAS cluster in at least one sample. The usage of an individual PAS was calculated as the number of its PAS reads divided by the sum of PAS reads in the gene. Genes that use only one identical PAS in all tissues sampled in this study were defined as single-PAS genes.

For allelic PAS usage, only reads unambiguously assigned to specific alleles were used. The allelic usage of an individual PAS was calculated as the number of allele-specific reads mapped to this PAS divided by the sum of all allele-specific PAS reads in the gene. Known imprinted genes in mouse extracted from the Geneimprint database (http://www.gene-imprint.com/site/genes-by-species) and genes on the sex chromosomes or mitochondria were excluded from allelic PAS analyses. In order to avoid inaccurate calculation of allelic PAS usage due to sampling errors for low coverage PAS regions, genes with the sum of allele-specific PAS reads from either allele < 10 in one sample were removed. For the remaining PASs, we further estimated their relative PAS usages by combining allelic reads from both alleles and compared the result to the usage estimated using both allelic reads and common reads. If the difference between the two estimates was > 10%, we regarded the PAS cluster as having insufficient coverage of unambiguously assigned allelic PAS reads for accurate estimation of allelic PAS usage, and it was therefore not used in further allelic analyses (Xiao et al, 2016). We compared the allelic PAS usage divergence between the two alleles using DEXSeq (Anders et al, 2012). Only PASs with Benjamini–Hochberg-adjusted P value < 0.05 and average allelic difference in PAS usage > 10% were defined as divergent.

## Estimation of gene expression level

Since PAS reads were derived from the 3′ end of polyA RNA, one read from each transcript, normalization for transcript length, as often performed for regular RNA-seq data, is not needed for quantifying gene expression. Therefore, the expression level of a gene was calculated as the sum of total PAS reads in the gene divided by sum of PAS reads from all the genes in the sample and multiplied by $10^6$ (referred to as reads per million mapped reads, "RPM"). Allelic gene expression was calculated as the sum of allele-specific PAS reads from the gene divided by the sum of total PAS reads from all the Refseq genes in the sample and multiplied by $10^6$. Differential gene expression between alleles was analyzed with DESeq2 (Love et al, 2014). Genes were considered as divergently expressed when the fold change between alleles was > 2 and the FDR < 0.05.

## Down-sampling of sequencing data

In order to avoid potential sampling bias in PAS identification caused by differences in sequencing depth between genes of different expression level, we generated a down-sampled dataset in which each gene was covered by the same sequencing depth. In brief, we first pooled the reads from all the sequenced samples after

normalizing them to the same sequencing depth and then randomly picked 100 reads mapped to each gene, to determine the number of used PASs and their relative usage.

### APA diversity within tissues, variability between replicates, and APA change across tissues

To quantify the APA diversity within a tissue, we used the Shannon index (Shannon, 1948) defined as $\text{Shannon index} = -\sum_i^n \log(p_i)p_i$ where $p_i$ is the usage of $i$th PAS usage in a gene. The Shannon index measures the entropy of the alternative PAS usage, which reflects both the number of different PASs and the evenness of the usage distribution across these sites.

To measure the PAS variability between biological replicates, we calculated for each gene in each replicate a vector containing the usage of all its PASs ($v_{j,k} = [\text{PAS}_1, \text{PAS}_2, \ldots \text{PAS}_i, \ldots \text{PAS}_n]$, where $j$ and $k$ indicate gene $j$ in the $k$th replicate in a tissue and $i$ indicates the $i$th PAS in a gene) and used the vectors' maximum norm difference between three replicates as APA replicate variability ($var_j = max\ ([|v_{j,1}\text{-}v_{j,2}|,\ |v_{j,2}\text{-}v_{j,3}|,\ |v_{j,3}\text{-}v_{j,1}|])$. In order to remove the potential biases introduced by different PAS number and reads depth, we created a mock replicate dataset by pooling all reads from three biological replicates and randomly assigning the reads into three groups where a gene's total read number is equal to the original read number. The mock dataset was then used to calculate the mock APA replicate variability. We subtracted the average variability of 100 randomly simulated mock replicates from the original replicate variability for each gene $j$ as adjusted variability ($\text{adjust var}_j = var_j - \sum_{l=1}^{100} var_{\text{mock}j,l}/100$).

Between tissues, we calculated for each gene in each tissue a vector containing the usage of all its PASs and used the vectors' maximum norm difference between a tissue pair as APA difference. The mock data were created by pooling the dataset from two tissues and randomly assigning them back to the two tissues and were used for calculating the adjusted APA difference similarly as above. We defined the gene's switch score as the maximum adjusted APA difference between any two tissues where the gene is expressed.

### Genes with brain-specific minor functional PASs and genes with exactly three PASs

Genes with brain-specific minor functional PASs were selected from genes which were classified into the F2 group exclusively in cortex and/or cerebellum (their rank 2 PAS usages in the other seven tissues were not high enough to be considered as genes with potentially functional minor PASs in these tissues).

To clarify different features between first, middle, and last PAS in 3′UTR, we defined a gene with exactly three PASs in 3′UTR if (i) it has exactly three PASs located in 3′UTR, and (ii) the sum of its three 3′UTR PASs usages is above 50% in every tissue where the gene is expressed.

### Estimation of sequence conservation, sequence variant density, and dN/dS ratios

PhastCons scores of the *Glires* clade (rodents, rabbits, etc.) were used to estimate sequence conservation. The PhastCons score data were obtained by PHAST (Siepel & Haussler, 2005; http://compge n.cshl.edu/phast/phastCons-tutorial.php). Sequence variant density between C57BL/6J and SPRET/EIJ was calculated in the chain file as performed in our previous F1 hybrid study (Gao *et al*, 2015). dN/dS ratios (ratio of the number of non-synonymous substitutions per non-synonymous site to the number of synonymous substitutions per synonymous site) between the house mouse (*M. musculus*) and rat (*Rattus norvegicus*) were downloaded from the Ensembl database (ensembl.org), to estimate selective constraints on the amino acid sequences of proteins.

### Estimation of C-terminal peptide conservation

Among the genes in which the dominant and the 2nd most used PASs are conserved between humans and mouse, those with alternative last exons were selected. The C-terminal peptides of the two isoforms of the gene were obtained from the mm10 annotated genome dataset (http://ensemblgenomes.org/). The conservation score of C-terminal peptides between humans and mouse was calculated as the peptide alignment score divided by mean peptide length, using the Blosum62 scoring matrix (Pearson, 2013) and the default penalties for gap opening ($-11$) and extension ($-1$). The C-terminal peptides were considered as conserved if their conservation scores are positive.

### MicroRNA target site density

We used conserved microRNA target sites annotated in TargetScanMouse (Agarwal *et al*, 2015; http://www.targetscan.org/mmu_72) to calculate the density of microRNA target sites in different 3′UTR regions. For a gene with its dominant PAS and rank 2 PAS in 3′UTR, we selected 1,000-nt upstream region of the distal PAS or the whole region between dominant PAS and rank 2 PAS if the region length is < 1,000 nt. The microRNA target site density in each group was calculated as the average microRNA target number at each position of the regions. The microRNA densities of the regions between the major PASs and 2nd PASs located in the 3′UTR were calculated in the same way. For the 300-nt upstream region of a PAS, if the region overlaps any coding regions annotated in Refseq, the region was shortened to its non-overlapping 3′UTR. The microRNA target site density of this region was calculated similarly as above.

### Clustering of allelic APA as well as gene expression pattern across multiple tissues

For APA pattern and mRNA abundance clustering, the 5,264 genes with at least one PAS having sufficient allelic PAS reads in all nine tissues were used. For each of these genes, the PAS with maximum average usage was used for APA pattern clustering. Hierarchical clustering was performed using Pearson correlations as a similarity measure.

## Data availability

All the sequencing data generated from this study have been submitted to the NCBI under the accession number PRJNA587365 (https://www.ncbi.nlm.nih.gov/bioproject/PRJNA587365).

**Expanded View** for this article is available online.

## Acknowledgements

This work was supported by the National Natural Science Foundation of China (Grant Nos. 31861133013, 31970601, and 31771443), the Shenzhen Science and Technology Program (Grant No.: KQTD20180411143432337), and the Science and Technology Innovation Commission of the Shenzhen Municipal Government (Grant Nos.: JCYJ20170307105752508 and JCYJ20180504165804015). Bioinformatics analysis was supported by the Center for Computational Science and Engineering of Southern University of Science and Technology. We thank Dr. Jean Jaubert and Dr. Xavier Montagutelli from the Pasteur Institute for providing F1 hybrid mice, and Dr. Claude Libert and Dr. Tino Hochepied from the Vlaams Instituut voor Biotechnologie (VIB) for F1 hybrid embryonic stem cells. Y.S. Li was supported by the Chinese Scholarship Council.

## Author contributions

WC conceived and designed the project. YL and MZ performed the sequencing experiments with help from WS, YH, YL, and LF. WL cultured the embryonic stem cells. YL and BS analyzed the data with help from XZ, FH, BZ, GL, JZ, and YH. YL, BS, and WC wrote the manuscript. All authors read and approved the final manuscript.

## Conflict of interest

The authors declare that they have no conflict of interest.

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
