## [Review Process File · Molecular Systems Biology]

Pan-tissue analysis of allelic alternative polyadenylation suggests widespread functional regulation

Yisheng Li, Bernhard Schaefer, Xudong Zou, Min Zhang, Florian Heyd, Wei Sun, Bin Zhang, Guipeng Li, Weizheng Liang, Yuhao He, Juexiao Zhou, Yunfei Li, Liang Fang, Yuhui Hu, and Wei Chen.

Review timeline:

Submission date:	18 th November 2019
Editorial Decision:	20 th December 2019
Revision received:	28 th January 2020
Editorial Decision:	25 th February 2020
Revision received:	29 th February 2020
Accepted:	11 th March 2020

Editor: Maria Polychronidou

Transaction Report:

1st Editorial Decision

20th December 2019

Thank you again for submitting your work to Molecular Systems Biology. We have now heard back from the three referees who agreed to evaluate your study. Overall, the reviewers acknowledge that while the conceptual advance seems modest, the study seems carefully performed and is likely to be relevant for the APA field. They raise however a series of concerns, which we would ask you to address in a revision.

As you will see below, the most substantial concern is raised by reviewer #2 who is not convinced that the study in its current form represents a decisive advance. Reviewer #2 recommends extending the study by including further analyses on the functional relevance of genes that undergo tissue-specific APA, in order to enhance the conceptual novelty and the level of biological insight provided by the study. We think that addressing this concern is important and would indeed significantly strengthen the manuscript.

REFeree REPORTS

Reviewer #1:

This manuscript by Li et al. uses 3' end sequencing to examine alternative polyadenylation (APA) isoform expression in multiple tissues from F1 hybrid of two mouse inbred strains. The authors identified groups of genes with respect to APA isoform expression levels, diversity, and regulation across tissues. A big part of this work is to revisit the conclusion made by a recent paper from Xu

and Zhang, which posits that APA is largely deleterious and not adaptive. The results of the current work indicate that the Xu and Zhang's theory applies only to certain genes. There are many genes that do not follow the theory. Overall, this is a nicely carried out study, and the conclusions should be of value to researchers studying APA.

1. Some aspects, however, are not clearly described and read a bit convoluted. For example, the authors should indicate in the Abstract with clarity what fractions of genes fall into the so-called APA error type and the APA functional type, respectively. It may also be a good idea to show their experimental design in Figure 1.
2. The authors need to examine polyA site conservation across species for APA error type vs. APA functional type, similar to what the Tian lab did recently for human to mouse comparisons. This would additionally indicate whether polyA site conservation correlates with functions.

Reviewer #2:

The paper by Li et al. is an exhaustive tissue specific APA analysis using two closely related mouse species. The APA data is meticulously interrogated with the aim to separate functionally relevant APA events from molecular noise. They identify around 4000 genes that undergo APA and may be of functional importance. They then show that genes with potentially functional minor polyadenylation sites result in APA transcripts that may be tissue specifically regulated by miRNAs and more specifically in neuronal tissue may be associated subcellular localisation. They then propose that the alternative usage of PAS is regulated by cis-element, transcriptionally and competition between different PAS.

Overall, I find the authors address the very pressing issue of identifying relevant and functionally important APA. Their approach is systematic, logical and uses appropriate pipelines and the data largely supports the conclusions made. I have no major technical issues with the paper. However as detailed below I am not convinced that the findings in their current form represent a major advance for the APA field.

Major points:

My main issue with the paper is that I struggle to extract its key message and I am somewhat left in limbo as to what the actual major new findings are. Their results show that whilst APA diversity in many genes can be attributed to molecular errors, they can extract events that are likely to be of functional importance. This is interesting but does not represent a major advance from the Xu and Zhang paper where the concept of erroneous APA was first introduced. In addition, their other major conclusions that the usage of PAS is not just regulated by cis elements but includes coupling to transcription and competition between sites is not a surprising finding. This work generated a large and interesting tissue and species specific data set that has the potential to identify and characterise highly relevant APA. I believe the manuscript would be significantly strengthened by a detailed analysis of the function and physiological importance of the isolated genes that undergo tissue specific APA in their system.

Reviewer #3:

In this manuscript, Li et al. performed a detailed analysis of the usage of polyadenylation sites (PAS), their conservation and relationship with expression level. For that they use tissue and allele specific polyadenylation specific RNA-Seq datasets that they generate. The underlying question of the study is up to what degree there are evolutionary evidences for alternative polyadenylation site usage being of functional relevance, or if they are mainly caused by molecular errors as proposed recently by Xu and Zhang (Cell Syst 2018). They confirm some of the previous observations from that study. For example, that for those genes that use mainly one polyadenylation site (>90%), the alternative polyadenylation sites are likely associated to molecular errors. However, in many other cases they show evidences of general functional usage (not molecular errors). This is particularly clear in those cases where there is not only one major PAS or when the PAS usage is different across tissues.

This work is thus an expansion and rebuttal to the previous work from Xu and Zhang (Cell syst 2018). The value of this manuscript is to show that when considering in more detail the intricacies of the polyadenylation process, the same analysis leads to a different conclusion. This is a general question in biology, and although it is a question that likely does not have a simple answer, in my opinion Li et al present a more balanced view of the topic.

Minor:

1. I was not able to find any access code for the deposited datasets (e.g. GEO, Arrayexpress...). The data should be made easily accessible prior to acceptance.

2. dN/dS should be defined the first time it is used.

1st Revision - authors' response

28th January 2020

Responses to Reviewers' comments

We thank the reviewers for the valuable comments, which were very helpful for revising our manuscript. We have tried our best to respond to the comments. Below are our point-to-point responses.

Reviewer #1:

1. Some aspects, however, are not clearly described and read a bit convoluted. For example, the authors should indicate in the Abstract with clarity what fractions of genes fall into the so-called APA error type and the APA functional type, respectively. It may also be a good idea to show their experimental design in Figure 1.

Answer: Thank you for pointing this out. We added a schema outlining our experimental design and analysis pipeline as figure 1.

We modified the abstract (page 2, lines 30-35 in the original manuscript, page 2, lines 28-32 in the revised version) to indicate the fractions of APA functional type (the union of F2 and switched genes) and APA error type (identical to the F1 group with dominant PAS usage above 90% in all expressing tissues) multi-PAS genes as follows (changes indicated in yellow):

“We found a clearly negative correlation between gene expression and APA diversity for the 2,866 genes (24.9%) with a dominant polyadenylation site (PAS) usage above 90%, suggesting that their other PASs represent molecular errors.

Among the remaining genes with multiple PASs, 3,971 genes (34.5%) express two or more isoforms with potentially functional importance.

2. The authors need to examine polyA site conservation across species for APA error type vs. APA functional type, similar to what the Tian lab did recently for human to mouse comparisons. This would additionally indicate whether polyA site conservation correlates with functions.

Answer: Thank you for this valuable suggestion. We utilized the cleavage and polyadenylation site information for human and mouse deposited in the polyA DB database and compared the “APA error type” (F1) and “APA functional type” genes for their PAS conservation. We found that “APA functional type” genes indeed had larger numbers of conserved minor PASs than F1 genes, and added the following sentences to our manuscript (page 15, lines 339-348 in the revised version):

“As shown above, a large fraction of F2 genes and switch genes appears to have two or more functional PASs and the two groups overlap to a significant degree. Therefore, we defined their union (comprising 3,971 genes) as “APA functional type” genes. For F1 genes in contrast minor PAS usage most likely reflects molecular error.

We utilized the cleavage and polyadenylation site information for human and mouse deposited in the polyA DB database (Wang et al. 2018b) and compared the F1 and “APA functional type” genes for their PAS conservation. We found that genes with two or more potentially functional PASs indeed had larger numbers of conserved minor PASs than F1 genes: whereas 2,443 (61.52 %) “APA functional type” genes had two or more conserved PASs with human, this was the case only for 853 (29.77 %) F1 genes (Fig EV5A).”

Reviewer #2:

However as detailed below I am not convinced that the findings in their current form represent a major advance for the APA field.

Major points:

My main issue with the paper is that I struggle to extract its key message and I am somewhat left in limbo as to what the actual major new findings are. Their results show that whilst APA diversity in many genes can be attributed to molecular errors, they can extract events that are likely to be of functional importance. This is interesting but does not represent a major advance from the Xu and Zhang paper where the concept of erroneous APA was first introduced. In addition, their other major conclusions that the usage of PAS is not just regulated by cis elements but includes coupling to transcription and competition between sites is not a surprising finding.

Answer:

While we agree that competition between PASs and coupling between transcriptional regulation and APA are not counter-intuitive results, they constitute novel findings of significant importance for the interpretation of APA diversity across tissues and between species, because both challenge the notion that a correlation between low expression level and high molecular diversity or allelic divergence can be simply interpreted as the result of relaxed selective constraints. The scaling effect resulting from molecular competition between different sites has only recently been described for alternative splicing. Our finding of a similar effect in APA suggests that differences of larger magnitude between alleles are not necessarily due to the accumulation of *cis*-regulatory mutations but can result from the interaction of molecular competition with small regulatory changes. Similarly, the relationship between major PAS location, distal PAS usage and gene expression level across tissues has not been shown before, and it suggests that the *trans*-regulatory machinery has evolved in a way that a reduced transcription rate may automatically lead to reduced polyadenylation accuracy, independent of *cis*-regulatory mutations. These conclusions help to resolve the controversy regarding the functional importance of transcript isoform diversity and contribute constructively to the reemerging “neutralist-selectionist” debate with regards to molecular phenotypes. They place Xu’s and Zhang’s error hypothesis in a new

conceptual framework, where increased error might not result from neutral drift in regulatory elements but 1. from the kinetics of competition (scaling effect) and 2. from natural selection favoring a coupling of gene regulatory layers in a way that assures the allocation of limited resources towards highly expressed genes (coupling between transcription and APA).

To clarify these points, we extended the first paragraph of the Discussion as follows (added passages in yellow; original manuscript: page 18, lines 412-420; revised version: page 20, lines 462-479):

“APA is a common process in higher eukaryotes which has received increased attention in recent years with regards to its function, regulation, and evolution. Improved high-throughput deep sequencing technology has enabled researchers to identify a growing number of genes undergoing APA, and it is often assumed, that the detected variation in PAS usage is biologically relevant. However, cases where functions of different APA isoforms have been clearly demonstrated are still rare compared to the ubiquity of the phenomenon. It has therefore been proposed that most observed APA is noise (“error hypothesis”) (Xu & Zhang, 2018) and that APA divergence between species is largely neutral or slightly deleterious (“neutral theory”) (Zhang, 2018). Here we address these issues in an F1 mouse model by investigating APA diversity within each tissue, APA differences between tissues and *cis*-regulatory divergence between *Mus musculus* (C57BL/6J) and its sister species *Mus spretus* (SPRET/EiJ). We show that the error hypothesis does not predict the differences between single-PAS and multi-PAS genes and that it only applies to a specific subset of multi-PAS genes. Furthermore, we demonstrate that higher APA diversity or divergence can only partially be attributed to a relaxation of selective constraint on lowly expressed genes, whereas functional coupling between transcriptional and APA regulation and competition between different PASs constitute additional factors with a significant impact on APA variation. Finally, we provide a set of genes with likely functional APA.”

This work generated a large and interesting tissue and species specific data set that has the potential to identify and characterize highly relevant APA. I believe the manuscript would be significantly strengthened by a detailed analysis of the

function and physiological importance of the isolated genes that undergo tissue specific APA in their system.

Answer:

Thank you for this valuable suggestion. As detailed below, we conducted additional analyses to identify high-confidence candidates for functional APA and tissue-dependent regulation and provide an extensive supplementary data set to facilitate the prioritization of future mechanistic studies on APA function.

Among the combined set of F2 and switched genes (APA functional type as defined above) we compiled a subset of high-confidence candidate genes for functional APA by selecting those with comparatively high expression levels in at least one tissue with high usage of the potentially functional minor PAS (see newly added paragraph below). We added a table containing the information for these high-confidence candidate genes with potentially two or more functional PASs, indicating all tissues where a minor PAS is putatively functional and type of APA (Multi_PAS_genes.xlsx).

Additionally, we show that in “functional type” genes with APA involving alternative splicing of the last exon and with the two PASs conserved between mouse and human, both exons encode highly conserved amino acid sequences, further indicating their functional importance. Interestingly, the mRNA isoforms produced by the 2nd PAS of APA functional type genes tend to have more conserved C-terminal peptides than those of other gene groups (Fig. EV5A). We provide a detailed table of these genes (Alternative_Last_Exon.xlsx) and two examples where the importance of different isoforms has been demonstrated in previous studies.

We added the following passages in the text:

In Results (page 15-17, lines 349-386 in the revised manuscript):

“To compile a high-confidence set of candidate genes with functional APA more stringently we selected all genes from the “APA functional type” group which either showed a positive correlation across tissues between gene expression level and usage of a potentially functional minor PAS (secondary PAS) or have an expression level higher than the median of all expressed genes in the tissue with the highest usage of the secondary PAS. In total, 2,218 genes fell into this category. Among these, 1,571 (70.83%) contain two or more conserved PASs with *Homo sapiens* and might therefore be of functional relevance also in humans.

Next, we were interested in finding candidates for functional APA affecting the coding region. As shown in our results, in many cases the usage of PASs upstream of the last stop codon can be attributed to molecular error. This might be different if APA is coupled with alternative splicing of the last exon, as recent evidence shows the functional importance of different protein-coding isoforms produced in this way and their differential localization in neurons (Taliaferro et al. 2016). An additional indication of functional APA is evolutionary conservation of the PASs involved. Among the genes with both the dominant and the second PASs conserved between human and mouse we found in total 171 genes with APA involving alternative last exon splicing. Among the eight F1 genes in this group, only one contains conserved C-terminal peptides specifically encoded by the minor isoforms (Fig EV5B). In contrast, 102 out of 121 genes with potentially functional APA exhibit conserved amino acid sequences encoded specifically by the minor isoforms between human and mouse (76 out of the 102 genes are high-confidence candidates; see extended table Alternative_Last_Exon.xlsx for the detailed information). For two of these genes, *Lmna* and *Cdc42*, both isoforms have been shown to exhibit distinct function in previous studies. The *Lmna* gene encodes protein lamin, which functions in nuclear mechanics and genome regulation (Gruenbaum & Foisner, 2015). As shown in Fig EV5C, *Lmna* gene expressed two APA isoforms in all the eight tissues as well as the ES cells. Compared to the long isoform lamin-A, the shorter isoform (lamin-C) lacks the amino acid residues encoded by exons 11 and 12. It has been shown that the short isoform lacks the C-terminal isoprenylation motif (CaaX motif) and therefore, in contrast to the longer lamin-A isoform, cannot be lipidated (Fisher, Chaudhary, & Blobel, 1986). Importantly, the lamin-C-only-expressing mice were reported to have reduced mitochondrial activity and energy

expenditure as well as increased fat mass and lifespan, demonstrating the functional importance of both APA isoforms (Lopez-Mejia et al., 2014). The *Cdc42* gene encodes a small GTPase of the Rho family which regulates cellular morphology and neurite outgrowth in the brain (Kang et al., 2008). It produces a brain-specific isoform ended with exon 6 (Cdc42E6) and a ubiquitous isoform ended with exon 7 (Cdc42E7) (Fig EV5D). It has been shown that the two isoforms were with distinct subcellular localization pattern in neurons determined by their different 3' UTRs (Mattioli et al., 2019). Even more, Cdc42E6 can promote spine induction more robustly than Cdc42E7 and is responsible for activity-driven synapse remodeling (Kang et al., 2008). "

In Methods (page 30, line 720-728): "Among the genes in which the dominant and the 2nd most used PASs are conserved between human and mouse, those with alternative last exons were selected. The C-terminal peptides of the two isoforms of the gene were obtained from the mm10 annotated genome dataset (<http://ensemblgenomes.org/>). The conservation score of C-terminal peptides between human and mouse was calculated as the peptide alignment score divided by the mean peptide length, using the Blosum62 scoring matrix (Pearson, 2013) and the default penalties for gap opening (-11) and extension (-1). The C-terminal peptides were considered as conserved if their conservation scores are positive."

Reviewer #3:

1. I was not able to find any access code for the deposited datasets (e.g. GEO, Arrayexpress...). The data should be made easily accessible prior to acceptance.

Answer: In addition to the raw data which are accessible under link <https://dataview.ncbi.nlm.nih.gov/object/PRJNA587365?reviewer=3ok6ov1tpc80h03jq73n5oh3ld> we uploaded summarized data of the PAS usage, gene expression

level, PAS features (type, adjusted variability and dN/dS ratio of the corresponding gene) for all multi-PAS genes in the expanded tables (Multi_PAS_genes.xlsx).

2. dN/dS should be defined the first time it is used.

Answer: Thank you for pointing this out. We added a definition in the main text (page 7, lines 141-144, revised version):

“In addition, we examined constraints on protein sequence evolution, measured by the ratio of the rate of nonsynonymous substitution to the rate of synonymous substitution (dN/dS ratio), with lower ratios reflecting stronger purifying selection on amino acid sequence (Miyata & Yasunaga, 1980). We found that ...”

Reference

- Fisher, D. Z., Chaudhary, N., & Blobel, G. (1986). cDNA sequencing of nuclear lamins A and C reveals primary and secondary structural homology to intermediate filament proteins. *Proceedings of the National Academy of Sciences of the United States of America*.
<https://doi.org/10.1073/pnas.83.17.6450>
- Gruenbaum, Y., & Foisner, R. (2015). Lamins: Nuclear Intermediate Filament Proteins with Fundamental Functions in Nuclear Mechanics and Genome Regulation. *Annual Review of Biochemistry*. <https://doi.org/10.1146/annurev-biochem-060614-034115>
- Kamieniarz-Gdula, K., & Proudfoot, N. J. (2019). Transcriptional Control by Premature Termination: A Forgotten Mechanism. *Trends in Genetics*.
<https://doi.org/10.1016/j.tig.2019.05.005>
- Kang, R., Wan, J., Arstikaitis, P., Takahashi, H., Huang, K., Bailey, A. O., ... El-Husseini, A. (2008). Neural palmitoyl-proteomics reveals dynamic synaptic palmitoylation. *Nature*. <https://doi.org/10.1038/nature07605>
- Lopez-Mejia, I. C., De Toledo, M., Chavey, C., Lapasset, L., Cavelier, P., Lopez-Herrera, C., ... Tazi, J. (2014). Antagonistic functions of LMNA isoforms in energy expenditure and lifespan. *EMBO Reports*.
<https://doi.org/10.1002/embr.201338126>
- Mattioli, C. C., Rom, A., Franke, V., Imami, K., Arrey, G., Terne, M., ... Chekulaeva, M. (2019). Alternative 3 UTRs direct localization of functionally diverse protein isoforms in neuronal compartments. *Nucleic Acids Research*.
<https://doi.org/10.1093/nar/gky1270>
- Miyata, T., & Yasunaga, T. (1980). Molecular evolution of mRNA: A method for estimating evolutionary rates of synonymous and amino acid substitutions from homologous nucleotide sequences and its application. *Journal of Molecular Evolution*. <https://doi.org/10.1007/BF01732067>
- Pearson, W. R. (2013). Selecting the right similarity-scoring matrix. *Current Protocols in Bioinformatics*. <https://doi.org/10.1002/0471250953.bi0305s43>
- Taliaferro, J. M., Vidaki, M., Oliveira, R., Olson, S., Zhan, L., Saxena, T., ...

- Burge, C. B. (2016). Distal Alternative Last Exons Localize mRNAs to Neural Projections. *Molecular Cell*. <https://doi.org/10.1016/j.molcel.2016.01.020>
- Xu, C., & Zhang, J. (2018). Alternative Polyadenylation of Mammalian Transcripts Is Generally Deleterious, Not Adaptive. *Cell Systems*, 6(6), 734-742.e4. <https://doi.org/10.1016/j.cels.2018.05.007>
- Zhang, J. (2018). Neutral Theory and Phenotypic Evolution. *Molecular Biology and Evolution*, 35(6), 1327–1331. <https://doi.org/10.1093/molbev/msy065>

2nd Editorial Decision

25th February 2020

Thank you again for sending us your revised manuscript. We have now heard back from the referee who was asked to evaluate your study. As you will see below, the reviewer acknowledges that the study has improved as a result of the performed revisions. They mention however, that the study would benefit from text editing, in order to make the description of the different APA sub-group comparisons clearer and to make the main findings better accessible to a broad audience. We would ask you to perform these text changes in a revision.

REFeree REPORTS

Reviewer #2:

As I indicated in my original review, this is a comprehensive APA analysis but it is difficult to extract the key messages. The revised version is an improvement but the description of the multiple different APA sub-group comparisons still lacks clarity and makes this paper a very complicated read that will limit its reach.

2nd Revision - authors' response

29th February 2020

Responses to Reviewers' comments

Reviewer #2:

As I indicated in my original review, this is a comprehensive APA analysis but it is difficult to extract the key messages. The revised version is an improvement but the description of the multiple different APA sub-group comparisons still lacks clarity and makes this paper a very complicated read that will limit its reach.

Answer:

To clarify our findings, we added an additional figure (Fig EV5) to summarize our classification of genes. We modified our discussion and added the following

sentences to re-emphasize the fractions of APA functional type (the union of F2 and switched genes) and APA error type:

1. Original manuscript: page 20, lines 491-492; revised version: page 20, lines 470-480:

"Our results indicate that functional coupling between transcriptional and APA regulation (with higher expression levels leading to increased distal PAS usage) and competition between different PASs constitute additional factors with a significant impact on APA variation. Finally, we provide a set of genes with likely functional APA."

2. Original manuscript: page 21, lines 512-521; revised version: page 21-22, lines 501-510:

"When the relationship between gene expression levels and APA diversity within a tissue or APA changes across tissues is considered, the multi-PAS genes with major PAS located in 3' UTR can largely be separated into two distinct groups: 1. genes with dominant PAS usage above or equal to 90% within each tissue (F1 genes, Fig EV5), which show strong positive correlations between gene expression level and dominant PAS usage and 2. the remaining genes with more diverse APA, for which the negative correlations between gene expression and APA diversity become much weaker and even negligible (Fig 3C). This suggests that for the former group, all the other alternative PASs can be safely regarded as noise, while within the latter group, some genes might contain functional minor PASs (see discussion below, Fig EV5)."

3. Revised version: page 22, lines 512-513:

"In contrast to gene expression levels, dN/dS ratios, as an alternative measure of selective constraints, however, do not show any correlations with APA diversity/change."

4. Original manuscript: page 24, lines 576-586; revised version: page 24, lines 565-576:

" Finally, we showed **that** at least 3,778 multi-PAS genes **(F2 genes, Fig EV5)** express potentially functional minor PASs **simultaneously** in at least one tissue. Two lines of evidence support their functional relevance: 1. The conservation scores of the 300 nt upstream regions of these minor PASs are similar to those of major PASs, 2. The regions between the minor PASs and the major PASs in these genes are more conserved than in other genes and these regions contain higher densities of microRNA binding sites. Functional minor PASs **apparently are** particularly important in the brain, where the two isoforms are often localized in different subcellular compartments. A significant proportion of genes with two potentially functional PASs in the same tissue also show tissue-dependent regulation. Interestingly, switch-like regulation of different isoforms is also found in **193** FU genes **(Fig EV5)**, and in these genes the regions between the dominant and the rank 2 PAS have higher conservation scores and microRNA densities than in other FU genes **(Figs EV6C and D)**."

5. Revised version: page 24-25, lines 582-586:

" **Within the union of switch genes and F2 genes (3,971 genes in total), we define a subset of 2,218 high-confidence candidates for functional APA regulation with high gene expression levels in tissues with significant minor PAS usage. Among these 1,571 (70.83%) contain two or more conserved PASs with *Homo sapiens* and might therefore be of functional relevance also in humans.** "

3rd Editorial Decision

11th March 2020

Thank you for your prompt response and for performing the final requested changes. We are now satisfied with the modifications made and I am pleased to inform you that your paper has been accepted for publication.

Corresponding Author Name: Wei Chen

Manuscript Number: MSB-19-9367R